# In heart failure reactivation of RNA-binding proteins is associated with the expression of 1,523 fetal-specific isoforms

Matteo D'Antonio[1‡], Jennifer P. Nguyen[2,3‡], Timothy D. Arthur[3,4], Hiroko Matsui[5], Margaret K. R. Donovan[2,3], Agnieszka D'Antonio-Chronowska[1], Kelly A. Frazer[1,5]*

**1** Department of Pediatrics, University of California San Diego, La Jolla, California, United States of America, **2** Bioinformatics and Systems Biology Graduate Program, University of California, San Diego, La Jolla, California, United States of America, **3** Department of Biomedical Informatics, University of California, San Diego, La Jolla, California, United States of America, **4** Biomedical Sciences Graduate Program, University of California, San Diego, La Jolla, California, United States of America, **5** Institute of Genomic Medicine, University of California San Diego, La Jolla, California, United States of America

‡ These authors share first authorship on this work.
* kafrazer@health.ucsd.edu

**Data Availability Statement:** All data used in this manuscript is available through dbGaP, Gene Expression Omnibus (GEO), and Figshare. Bulk

## Abstract

Reactivation of fetal-specific genes and isoforms occurs during heart failure. However, the underlying molecular mechanisms and the extent to which the fetal program switch occurs remains unclear. Limitations hindering transcriptome-wide analyses of alternative splicing differences (i.e. isoform switching) in cardiovascular system (CVS) tissues between fetal, healthy adult and heart failure have included both cellular heterogeneity across bulk RNA-seq samples and limited availability of fetal tissue for research. To overcome these limitations, we have deconvoluted the cellular compositions of 996 RNA-seq samples representing heart failure, healthy adult (heart and arteria), and fetal-like (iPSC-derived car-diovascular progenitor cells) CVS tissues. Comparison of the expression profiles revealed that reactivation of fetal-specific RNA-binding proteins (RBPs), and the accompanied re-expression of 1,523 fetal-specific isoforms, contribute to the transcriptome differences between heart failure and healthy adult heart. Of note, isoforms for 20 different RBPs were among those that reverted in heart failure to the fetal-like expression pattern. We determined that, compared with adult-specific isoforms, fetal-specific isoforms encode proteins that tend to have more functions, are more likely to harbor RBP binding sites, have canonical sequences at their splice sites, and contain typical upstream polypyrimidine tracts. Our study suggests that compared with healthy adult, fetal cardiac tissue requires stricter tran-scriptional regulation, and that during heart failure reversion to this stricter transcriptional regulation occurs. Furthermore, we provide a resource of cardiac developmental stage-specific and heart failure-associated genes and isoforms, which are largely unexplored and can be exploited to investigate novel therapeutics for heart failure.

RNA-seq and genotype information are available for GTEx and iPSCORE through the dbGaP studies phs000424 (GTEx), phs000924 (iPSCORE, RNA-seq) and phs001325 (iPSCORE whole genome sequencing). scRNA-seq was obtained from dbGaP studies phs000924 (iPSCORE iPSC-CVPC) and phs001539 (adult heart). Heart failure bulk RNA-seq samples were obtained from GEO GSE46224. Supporting data for all figures and supplemental tables and R processing scripts are available on Figshare: https://urldefense.com/v3/__https://doi.org/10.6084/m9.figshare.13537343__;!!LLK065n_VXAQ!1R9Sum-bKT1Z8Se9q_8hgx-VFzQKlsT3mJw1LkXKJiM4XJR66sjn342BrxBGK9laIcc$.

**Funding:** This work was supported by a California Institute for Regenerative Medicine grant GC1R-06673-B, National Science Foundation - Civil, Mechanical and Manufacturing Innovation division award 1728497, and NIH grants HG008118 and HL107442 awarded to KAF. JPN, MKRD and TDA were supported in part by National Library of Medicine grant T15LM011271. TDA was supported in part by NIH grant F31HL158198. The funders had no role in study design, data collection and analysis, decision to publish, or preparation of the manuscript.

**Competing interests:** The authors have declared that no competing interests exist.

## Author summary

Heart failure is a chronic condition in which the heart does not pump enough blood. It has been shown that in heart failure, the adult heart reverts to a fetal-like metabolic state and oxygen consumption. Additionally, genes and isoforms that are expressed in the heart only during fetal development (i.e. not in the healthy adult heart) are turned on in heart failure. However, the underlying molecular mechanisms and the extent to which the switch to a fetal gene program occurs remains unclear. In this study, we initially characterized the differences between the fetal and adult heart transcriptomes (entire set of expressed genes and isoforms). We found that RNA binding proteins (RBPs), a family of genes that regulate multiple aspects of a transcript's maturation, including transcription, splicing and post-transcriptional modifications, play a central role in the differences between fetal and adult heart tissues. We observed that many RBPs that are only expressed in the heart during fetal development become reactivated in heart failure, resulting in the expression of 1,523 fetal-specific isoforms. These findings suggest that reactivation of fetal-specific RBPs in heart failure drives a transcriptome-wide switch to expression of fetal-specific isoforms; and hence that RBPs could potentially serve as novel therapeutic targets.

## Introduction

Heart failure is associated with increased expression of fetal-specific genes including cardiac ion channels [1], as well as fetal-specific isoforms of genes with pivotal roles in cardiac development, including *TTN* and *SCN5A* [2–4]. Multiple RNA binding proteins (RBPs), including *RBFOX2*, *HNRNPL*, *RBM24*, *CELF1*, *QKI*, *MBNL1* and *RBFOX1* [2,5–11], have cardiac developmental stage-specific expression. Since RBPs regulate alternative splicing outcomes (i.e. relative abundance of different alternatively splice isoforms), it has been hypothesized that RBPs are major drivers of transcriptomic diversity during cardiac development and between adult heathy and diseased states [12,13]. However, a transcriptome-wide analyses of RBP expression and alternative splicing differences in cardiovascular system (CVS) tissues between fetal and adult (healthy and diseased) stages, has yet to be conducted.

Until now, several limitations have hindered transcriptome-wide analyses of alternative splicing differences (i.e. isoform switching) between fetal and adult CVS tissues (heart and arteria). Adult CVS tissues are composed of multiple cell types, and different samples of the same tissue type are comprised of different proportions of these cell types [14,15]. Heterogeneity across bulk RNA-seq samples of the same tissue type limits the power to identify gene and isoform expression differences between different tissues and development stages. Furthermore, bulk RNA-seq experiments average gene expression across the population of cells in a given sample and thereby prevent the ability to capture isoform expression variability across different cell types. While in theory single cell transcriptomics may be more indicative than bulk RNA-seq for this type of analysis, most of the recently developed technologies fail to assess the full transcriptome, as they rely on the expression of only the 3' end of each transcript and, therefore, cannot be used to investigate isoform expression. Finally, the limited availability of human fetal heart tissues for research makes it impossible to obtain the large sample sizes required for the statistical power to perform transcriptome-wide analyses of the re-expression of fetal-specific isoforms. We and other have shown that induced pluripotent stem cell (iPSC) derived cardiovascular progenitor cells (iPSC-CVPC) show transcriptomic, epigenomic, morphological, structural and functional properties of their fetal counterparts,

including their cellular composition being a mixture of fetal-like cardiomyocytes and epicardial-derived cells (EPDCs) including myofibroblasts, vascular smooth muscle cells and endothelial cells, and can be used as a surrogate for fetal cardiac tissues for developmental, genetic and pharmacological studies [14,16–21].

To overcome the limitations of sample heterogeneity and isoform detection, we have taken advantage of 786 GTEx samples (heart: atrial appendage and left ventricle; arteria: aorta and coronary artery) collected from 352 adults [22] and 180 iPSCORE samples (iPSC-CVPCs) from 139 individuals [14]. Analyzing these 966 bulk RNA-seq samples representing CVS tissues at two developmental stages (fetal and adult) and three distinct types (iPSC-CVPC, heart and arteria), we found that the vast majority of RBPs are expressed at higher levels in the fetal-like iPSC-CVPC compared with adult CVS tissues and play a larger role regulating the expression of fetal-specific isoforms than adult-specific isoforms. We show that fetal-specific isoforms encode proteins that have more functions than their cognate adult-specific isoforms; and are also more likely to harbor RBP binding motifs, have canonical splice sites and contain typical upstream polypyrimidine tracts, suggesting that transcription is more strictly regulated in fetal than adult CVS tissues. To examine cell type-specific gene expression differences between fetal-like and adult CVS tissues, we performed cell type deconvolution of the 966 bulk RNA-seq samples. We validate the isoform switching of previously described cardiac markers, such as *SCN5A* [23], *TNNT2* [24], *ABLIM1* [25,26] and *TTN* [4], and show that the expression of thousands of genes and isoforms are associated with both CVS developmental stage and cell type.

To investigate the extent that the heart failure transcriptome reverts to a fetal-like program, we compared heart failure samples [20] with fetal-like iPSC-CVPC and adult healthy heart samples. We show that while the cellular composition of heart failure and healthy heart samples are similar, RBP genes as a class are overexpressed in heart failure, and more importantly, that the RBPs that are overexpressed substantially overlap those expressed at high levels in fetal-like iPSC-CVPC. Furthermore, we show that reactivation of fetal-specific RBPs in heart failure is associated with a transcriptome-wide isoform switch, which results in the expression of 1,523 fetal heart-specific isoforms. Our study provides a resource of cardiac developmental stage-specific and heart failure-associated genes and isoforms which the community will be able to exploit to investigate novel therapeutics for heart failure.

## Results

### RNA binding proteins show extensive expression differences between fetal-like and adult CVS tissues

To examine global transcriptome-wide gene expression differences between fetal-like iPSC-CVPC and adult CVS tissues we performed dimensionality reduction and clustering of the 966 bulk RNA-seq samples (180 iPSC-CVPCs, 227 aorta, 125 coronary arteries, 196 atrial appendages and 238 left ventricles; S1 Fig and S1 and S2 Tables). We observed that at intermediate-low (resolution = 0.01) samples divided into three clusters corresponding to iPSC-CVPC, "adult heart" (atrial appendage and left ventricle samples) and "adult arteria" (aorta and coronary artery samples, Figs 1A, 1B, S2A–S2E, S3 and S4 and S2 Table). The iPSC-CVPCs were associated with a higher cell division rate [27–29] and a lower pseudotime score (e.g. earlier developmental stage) [30], compared with the two adult CVS tissues (Fig S2F–S2H), which is congruent with the findings of earlier studies showing iPSC-CVPCs resemble fetal-like CVS tissue [14,16].

We next conducted pairwise gene expression comparisons of the three CVS tissues (fetal-like iPSC-CVPCs, adult heart, adult arteria) using linear regression and adjusting for possible

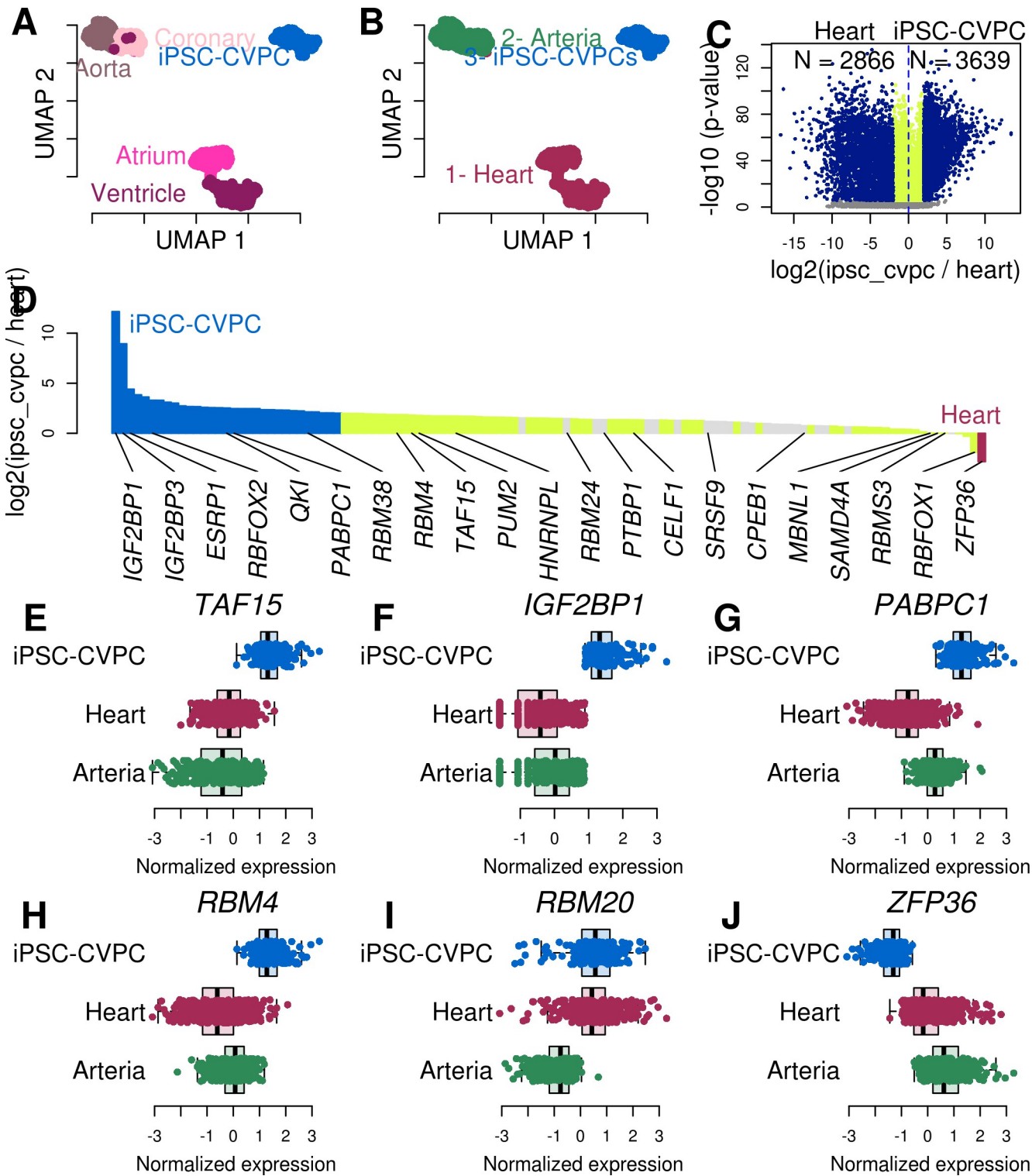

**Fig 1. RBPs are overexpressed in fetal-like CVS tissue.** (A, B) UMAP plots showing how the 966 RNA-seq samples cluster at varying resolutions: (A) shows the samples according to their tissue of origin (iPSC-CVPC, adult atrium, ventricle, aorta or coronary artery); In panel (B) samples are colored based on clustering at resolution = 0.01. S2A–S2E and S3 Figs show clustering at different resolutions. (C) Volcano plot showing differential gene expression between iPSC-CVPC and adult heart. The 6,505 genes with greater than four-fold difference ($log_2$ ratio > 2 or < -2) are shown in blue (FDR < 0.05) which include 2,866 genes with $log_2$ ratio < -2 (adult heart-specific) and 3,639 genes with $log_2$ ratio > 2 (iPSC-CVPC-specific). Genes that are significantly differentially expressed (FDR < 0.05) with $log_2$ ratio between -2 and 2 are in yellow. Genes that are not differentially expressed (FDR > 0.05) are shown in

gray. Volcano plots showing the differential expression between iPSC-CVPC and arteria, and between adult heart and arteria are shown in S5A and S5B Figs. (D) Barplot showing the log$_2$ ratio between the mean expression in iPSC-CVPC and adult heart for 122 RBPs with a known binding motif. iPSC-CVPC-specific RBPs are shown in blue, adult heart-specific RBPs are shown in red. All other differentially expressed RBPs (FDR < 0.05) are shown in yellow. The barplots showing the comparisons between iPSC-CVPC and adult arteria and between adult heart and arteria are shown in S5C and S5D Fig. The gene symbols of selected RBPs with cardiac- or development-specific functions are shown. (E-J) Boxplot showing gene expression levels (normalized TPM) in iPSC-CVPC, adult heart and adult arteria for six differentially expressed RBPs with cardiac functions. Other RBPs are shown in S6 Fig.

sources of technical bias and found 6,505 tissue-specific genes (31.9% of all expressed genes, absolute value of log$_2$ ratio > 2 and p-value < 0.05, F-test, after false discovery rate, FDR, correction using Bonferroni's method) between iPSC-CVPC and adult heart; 5,818 genes (28.5%) between iPSC-CVPC and adult arteria; and 3,358 genes (16.5%) between adult heart and adult arteria (Fig 1C, S5A and S5B and S3 Table). Because thousands of genes were differentially expressed between the iPSCORE fetal-like iPSC-CVPC and the GTEx adult CVS tissues, we investigated their function to confirm that our findings reflect biological differences, rather than technical artifacts. We performed functional enrichment analysis, which showed that genes overexpressed in both the adult heart and iPSC-CVPC compared with adult arteria were associated with cardiac muscle function, and that adult arteria-specific genes were enriched for immune response and processes associated with extracellular matrix organization (S4 Table). Genes with high expression levels in fetal-like iPSC-CVPC compared with both adult CVS tissues were strongly enriched for the RNA binding gene set (GO:0003723, p = 1.8 x 10$^{-57}$ and p = 5.0 x 10$^{-54}$, t-test, compared with adult heart and adult arteria, respectively). This gene set included 122 RNA binding proteins (RBPs) with known binding motifs [31–33], of which: 1) 93 were overexpressed in iPSC-CVPC compared with adult heart (FDR < 0.05), including 33 RBPs that were iPSC-CVPC-specific (log$_2$ ratio > 2 and FDR < 0.05); 2) four were overexpressed in adult heart compared with iPSC-CVPC (one adult heart-specific); and 3) 24 overexpressed in adult arteria compared with iPSC-CVPC (three adult arteria-specific, Figs 1D and S5C and S5D). Our functional enrichment analysis shows that iPSC-CVPC have more tissue-specific and overexpressed RBPs than either adult heart or adult arteria, and while there is a greater number of overexpressed RBPs in adult arteria compared with adult heart, the two adult CVS tissues have similar numbers of tissue-specific RBPs expressed, suggesting that RBP overexpression is a feature associated with fetal heart development.

We observed several RBPs known to be involved in fetal cardiac development (including *RBFOX2*, *HNRNPL*, *RBM24* and *CELF1*) and with adult functions (*MBNL1* and *RBFOX1*) [2,5–10] were respectively overexpressed in the iPSC-CVPC and adult CVS tissues (Figs 1D and S6). Additional RBPs overexpressed in iPSC-CVPCs included: *TAF15*, involved in cell proliferation [34] (Fig 1E and S3 Table); *IGF2BP1*, which regulates fetal hemoglobin alternative splicing [35] (Fig 1F); *PABPC1*, which regulates translation and cardiomyocyte growth [36] (Fig 1G); and *RBM4*, which regulates striated muscle differentiation [37] (Fig 1H). We also found *RBM20*, which has been shown to have rare inherited variants causing human dilated cardiomyopathy [38–40], to be overexpressed in both iPSC-CVPC and adult heart compared with adult arteria (respectively, p = 1.5 x 10$^{-30}$ and p = 1.3 x 10$^{-55}$, Fig 1I), consistent with its predominant expression in striated muscle [41]. The only RBP that was significantly overexpressed in both adult heart and adult arteria, compared with fetal-like iPSC-CVPC, was *ZFP36* (heart: log$_2$ ratio = -2.8, p = 3.6 x 10$^{-62}$; and arteria: log$_2$ ratio = -4.7, p = 4.1 x 10$^{-78}$, Fig 1J). Consistent with its adult-specific expression, this gene is involved in the inhibition of the immune and inflammatory responses highly expressed in atherosclerotic lesions, where it controls inflammatory response by inhibiting the expression of proinflammatory transcripts [42,43]. Overall, these analyses demonstrate that the vast majority of RBPs are expressed at higher levels in the fetal-like iPSC-CVPC compared with the adult CVS tissues, and suggest

that RBPs contribute to transcriptome-wide expression differences between the two developmental stages.

## Large-scale isoform switching occurs between fetal-like and adult CVS developmental stages

We hypothesized that the global expression differences of RBPs between developmental stages result in large-scale isoform switching between iPSC-CVPC, adult heart, and adult arteria. Indeed, across all expressed isoforms, we observed 19,270 differentially expressed isoforms between iPSC-CVPC and adult heart, including 1,534 iPSC-CVPC-specific and 3,043 adult heart-specific (Fig 2A and S5 Table). Likewise, between iPSC-CVPC and adult arteria, we found 1,785 iPSC-CVPC-specific isoforms and 3,742 adult arteria-specific isoforms, and between adult heart and adult arteria, we found 629 adult heart-specific isoforms and 319 adult arteria-specific isoforms. To confirm that the isoform usage differences between fetal-like iPSC-CVPC and adult heart and arteria were biologically relevant, we tested if we could capture the isoform switching of four cardiac genes known to have well-established developmental stage-specific isoforms (*SCN5A*, *TNNT2*, *ABLIM1* and *TTN*) and for all these genes, we found the expected associations between isoform expression and CVS developmental stage (S7 Fig). Interestingly, we discovered that 38 RBPs (31.1% of expressed RBPs) had isoforms with developmental stage-specific expression, including RBPs with known cardiac functions such as *MBNL1*, *FXR1*, *HNRNPM* and *FMR1* (Fig 2B–2E). Our analyses identified thousands of novel isoforms that undergo switching between fetal-like and adult CVS tissues and determined that approximately one-third of RBPs have isoforms that display CVS developmental stage-specific expression.

We hypothesized that differential isoform usage may contribute to the measurement of differential gene expression. Therefore, we tested if genes whose isoforms displayed differential usage between the fetal-like and adult CVS developmental stages were more likely to be differentially expressed themselves compared with genes whose isoforms did not show differential usage. We observed that genes that had differential isoform usage were 1.20 times more likely to be differentially expressed (p = 7.3 x $10^{-4}$, Fisher's exact test, iPSC-CVPC compared with adult heart, Fig 2A and S5 Table). These results suggest that a significant fraction of the measured differential gene expression is likely due to differences in isoform usage between developmental stages.

We performed functional enrichment analyses on genes that had at least one differentially expressed isoform between iPSC-CVPC and adult CVS tissues, and found they were enriched for regulating mRNA processing, including alternative splicing and RBP function (S6 Table). As a control, we examined genes with isoforms overexpressed in adult heart compared with adult arteria and found that they were enriched for cardiac muscle and mitochondrial functions, which represent known physiological differences between the two tissues. The overexpression of isoforms involved in mRNA processing in the fetal-like iPSC-CVPCs compared with adult CVS tissues suggests that there are transcriptome-wide regulatory differences between the two developmental stages.

## iPSC-CVPC-specific exons are enriched for functional protein domains

To further investigate the differences between CVS developmental stage-specific isoforms, we tested if the iPSC-CVPC-specific, adult heart-specific and arteria-specific isoforms had different associations with functional protein domains. We found that, although there was no enrichment for specific domains, iPSC-CVPC-specific exons were 1.21 more likely to encode for known protein domains [44] than adult heart-specific exons (p = 4.4 x $10^{-5}$, Fisher's exact

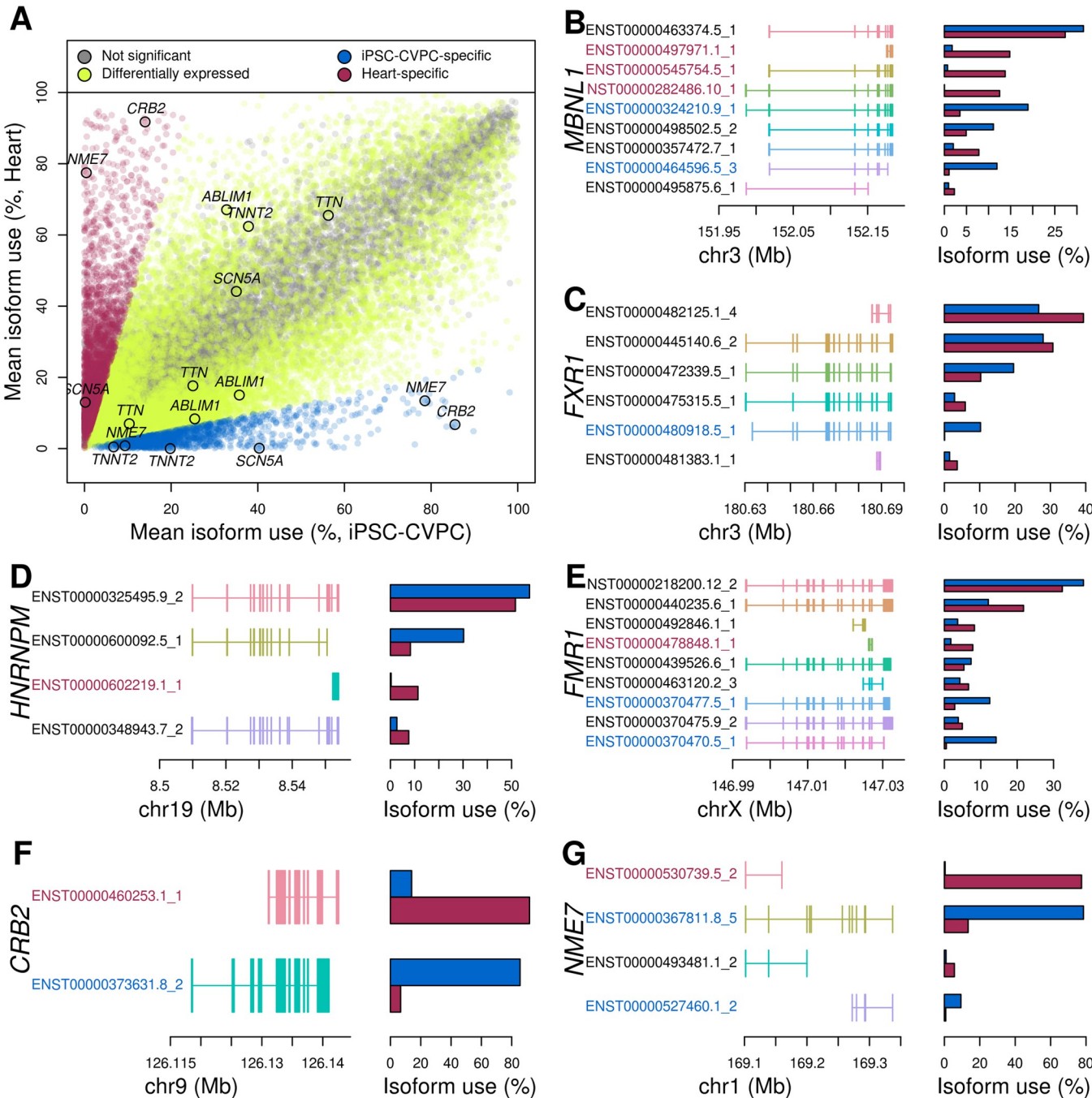

**Fig 2. Large-scale isoform switching occurs between fetal-like iPSC-CVPC and heart tissue.** (A) Scatter plot showing differential usage between adult heart and iPSC-CVPC for all expressed isoforms. iPSC-CVPC- and adult heart-specific isoforms are shown in blue and purple, respectively (absolute value of the log$_2$ ratio between mean isoform use between iPSC-CVPC and adult heart > 2 and FDR < 0.05). All other differentially expressed isoforms are shown in yellow (FDR < 0.05). Isoforms not differentially expressed are shown in gray. Isoform switching occurs between fetal-like iPSC-CVPC and adult heart for four RBPs (B-E) and two genes (F-G) with known cardiac functions. Left side of the figure represents the exon structure of each expressed isoform; the barplots on the right show the mean isoform use (%) of each isoform in iPSC-CVPC (blue) and adult heart (purple). Isoform IDs highlighted in blue are iPSC-CVPC-specific and isoform IDs highlighted in purple are adult heart-specific.

test) and 1.15 times than adult arteria-specific exons (p = 1.2 x $10^{-3}$, Fisher's exact test, S7 Table). We inspected the protein domains of novel developmental stage-specific isoforms in several genes with documented cardiac associated functions. *CRB2*, a gene involved in the differentiation of mesodermal cells [45], had one iPSC-CVPC-specific and one shorter adult-associated isoform (Fig 2F) lacking the three C-terminal exons encoding for its extracellular domain, suggesting that its function is likely impaired in adult cardiac tissues. *NME7*, which is involved in centrosome organization and mitotic spindle formation [46], had one iPSC-CVPC-specific, one iPSC-CVPC-associated and one adult heart-specific isoform (Fig 2G), which could underlie the significant differences in fetal and adult cardiac cell division rates [27]. Only the iPSC-CVPC-associated *NME7* isoform (ENST00000367811.8_5) was labeled by Gencode as "protein coding", whereas the adult heart-specific isoform is a processed transcript which lacks the N-terminal end and is likely not translated into a functional protein. Overall, these findings show that fetal-like iPSC-CVPC-specific isoforms were significantly more likely to overlap functional protein domains than their cognate adult-specific isoforms.

## iPSC-CVPC-specific exons are enriched for RBP binding and canonical splice sites

To investigate the association between differential isoform usage and differential RBP expression, we tested the overlap of CVS developmental stage-specific isoforms and experimentally determined (eCLIP) binding sites of 38 RBPs [31]. Of these 38 RBPs, 33 were overexpressed in iPSC-CVPC (including 12 iPSC-CVPC-specific) and five were not differentially expressed between iPSC-CVPC and adult heart. We observed that the genomic loci encoding iPSC-CVPC-specific isoforms were more likely to overlap RBP binding sites than the genomic loci encoding adult heart- or adult arteria-specific isoforms, whereas we did not observe significant differences between the two adult tissues (Figs 3A and 3B and S8 and S8 Table). We also investigated the occurrence of motifs for 122 RBPs (present in the GO:0003723 RNA binding gene set and have a known motif) in splice sites of CVS tissue-specific exons and found that the splice sites of iPSC-CVPC-specific exons were more likely to harbor RBP motifs than the splice sites of heart-specific and arteria-specific exons (S9 Fig and S9 Table). Conversely, splice sites for adult heart-specific and arteria-specific exons harbored similar numbers of RBP motifs. These results are consistent with our observation that the vast majority of RBPs are expressed at higher levels in the iPSC-CVPC and indicate that RBPs play a larger role controlling the expression of fetal-specific isoforms than adult-specific isoforms.

As alternative splicing outcomes are influenced by several factors including splice site strength, we further investigated the splice site sequences of CVS developmental-specific exons for differences in the frequency of canonical and non-canonical splice donor and acceptor sites between fetal-like and adult CVS tissues (Fig 3C–3I and S10 Table). At the splice donor site the first and fourth positions of iPSC-CVPC-specific exons were significantly enriched for the canonical nucleotides (G and A, respectively) compared with both adult heart and adult arteria-specific exons (respectively, p = 3.3 x $10^{-15}$ and p = 2.1 x $10^{-18}$ at the first position; p = 1.5 x $10^{-9}$ and p = 2.7 x $10^{-10}$ at the fourth position), whereas we did not observe significant differences between the two adult CVS tissues (p = 0.43 and p = 0.46, respectively for the first and fourth position). Since the genomic region immediately upstream of the splice acceptor usually contains a polypyrimidine tract (mostly thymine) of 15–20 nucleotides [47], we investigated if iPSC-CVPC-specific exons were enriched for having pyrimidines within the first 100 bp upstream of their splice acceptor site. Of the first 100 bps upstream of the splice acceptor sites, we found that 84 and 81 of the bps for the iPSC-CVPC-specific exons were significantly more likely to be thymine than adult heart-specific exons and adult arteria-specific exons,

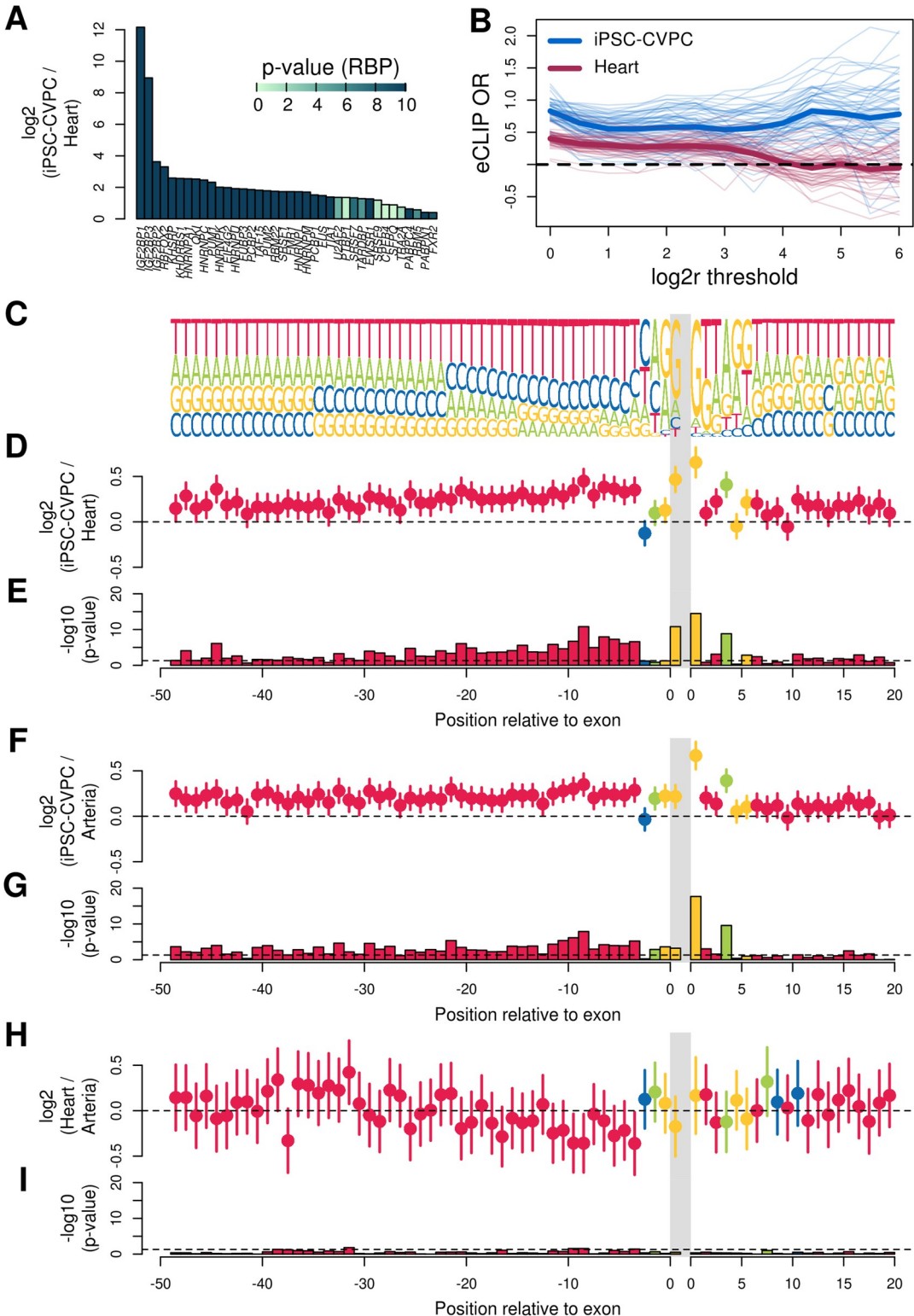

**Fig 3. iPSC-CVPC-specific exons are enriched for RBP binding and canonical splice sites.** (A) Barplot showing the differential expression of 38 RBPs with experimentally determined (eCLIP) binding sites between iPSC-CVPC and adult heart. Colors represent p-values (S3 Table). Each row represents one RBP. (B) Plot showing the enrichment of genomic loci encoding iPSC-CVPC-specific isoforms and adult-heart specific isoforms for overlapping experimentally determined (eCLIP) RBP binding sites. Each line corresponds to an eCLIP experiment. The X axis corresponds to log$_2$ ratio thresholds (mean

isoform usage in iPSC-CVPC/mean isoform usage in adult heart; FDR < 0.05). Enrichment was calculated by comparing the proportion of genes with differentially expressed isoforms passing the $\log_2$ ratio thresholds described on the X axis (S5 Table) and overlapping eCLIP peaks against the proportion of genes without any differentially expressed isoforms (FDR > 0.05 for all isoforms) overlapping eCLIP peaks. The thick lines represent the mean enrichments across all RBP experiments for iPSC-CVPC-specific isoforms (blue) or adult heart-specific isoforms (purple). The plot shows that, at all thresholds, iPSC-CVPC-specific isoforms are more likely to overlap eCLIP peaks than adult heart-specific isoforms. (C) Shown is the sequence logo of the 50 bp upstream of the splice acceptor site (left of gray bar) and the 20 bp downstream of the splice donor site (right of gray bar). Gray bar represents the location of the exon. Only the first nucleotide of the exon is shown. $\log_2$ ratio between the use of the most common nucleotide at each position between (D) iPSC-CVPC-specific and adult heart-specific exons, (F) iPSC-CVPC-specific and adult arteria-specific exons and (H) adult heart-specific and adult arteria-specific exons. Circles represent the estimate and segments represent the 95% confidence interval as calculated using the *fisher.test* function in R. (E, G, I) Barplots showing the $-\log_{10}$ (p-values, Fisher's exact test) of the enrichments shown in (D, F, H). The Y axis is the same across similar panels: in panel I, three positions have significant differences between adult heart and arteria, with p-values close to 0.05 (dotted line), whereas the majority of comparisons between iPSC-CVPC and each of the two adult tissues are significant in panels E and G. At each position, test results are shown only for the most common nucleotide. Each color represents the tested nucleotide, as shown in Panel C (A = green; C = blue; G = yellow; T = red). Enrichment results for all the nucleotides at all positions are shown in S10 Table.

respectively (FDR < 0.05, Fisher's exact test, S10 Table). Conversely, only eight positions were significantly different between adult heart-specific exons and adult arteria-specific exons. These analyses show that adult-specific exons are more likely to have non-canonical splice sites, consistent with previous studies showing that non-canonical splice sites tend to be tissue-specific and contribute to increasing proteome diversity between adult tissues [48,49]; on the other hand, iPSC-CVPC-specific exons are more likely to have canonical sequences at their splice sites and the typical upstream polypyrimidine tract, indicating that they are more likely to use conserved splicing machinery. Overall, the fact that fetal-specific isoforms being more likely to harbor RBP binding motifs, have canonical splice sites and contain typical upstream polypyrimidine tracts suggest that transcription may be more strictly controlled in fetal compared with adult CVS tissues.

## Estimated cell type proportions across 966 CVS samples

We hypothesized that the transcriptional differences between the fetal-like iPSC-CVPC and two adult CVS tissues were due to expression of developmental stage-specific genes and isoforms, while the transcriptional differences between the two adult CVS tissues were a consequence of their cellular composition differences. To test this hypothesis, we deconvoluted and determined the cell type composition [15] of each of the 966 bulk RNA-seq samples. Due to the sparsity of human scRNA-seq data, we leveraged the *Tabula Muris* resource [50], which contains deep scRNA-seq data from mouse heart and aorta that enables the accurate estimation of cell type proportions in human bulk RNA-seq samples [15]. The expression levels of marker genes for eight cardiac cell types (S10 Fig) was used as input into CIBERSORT [51] to estimate cell type composition (S11 and S12 Tables). The most common cell type in iPSC-CVPC and adult heart samples was cardiac muscle, whereas adult arteria samples were mainly composed of smooth muscle (Fig 4A). There were two cell types (fibroblasts and immune cells) present in the adult CVS tissues but not in the iPSC-CVPCs (mean proportion = 0.07% and 0.3%, respectively). A principal component analysis showed that expression heterogeneity across the 966 bulk RNA-seq samples was largely due to developmental stage (fetal-like versus adult) and cellular composition (S11 Fig). To validate the CIBERSORT-estimated cell type proportions we conducted a linear regression analysis between cell type proportions and gene expression levels (Figs 4A and S12 and S13 Table) and observed a strong positive correlation between the effect sizes for each gene and its expression in each cell type from two scRNA-seq CVS studies [14,52] (S13 Fig). Additionally, we performed a gene set enrichment analysis on the effect sizes of each cell type proportion across all genes, which

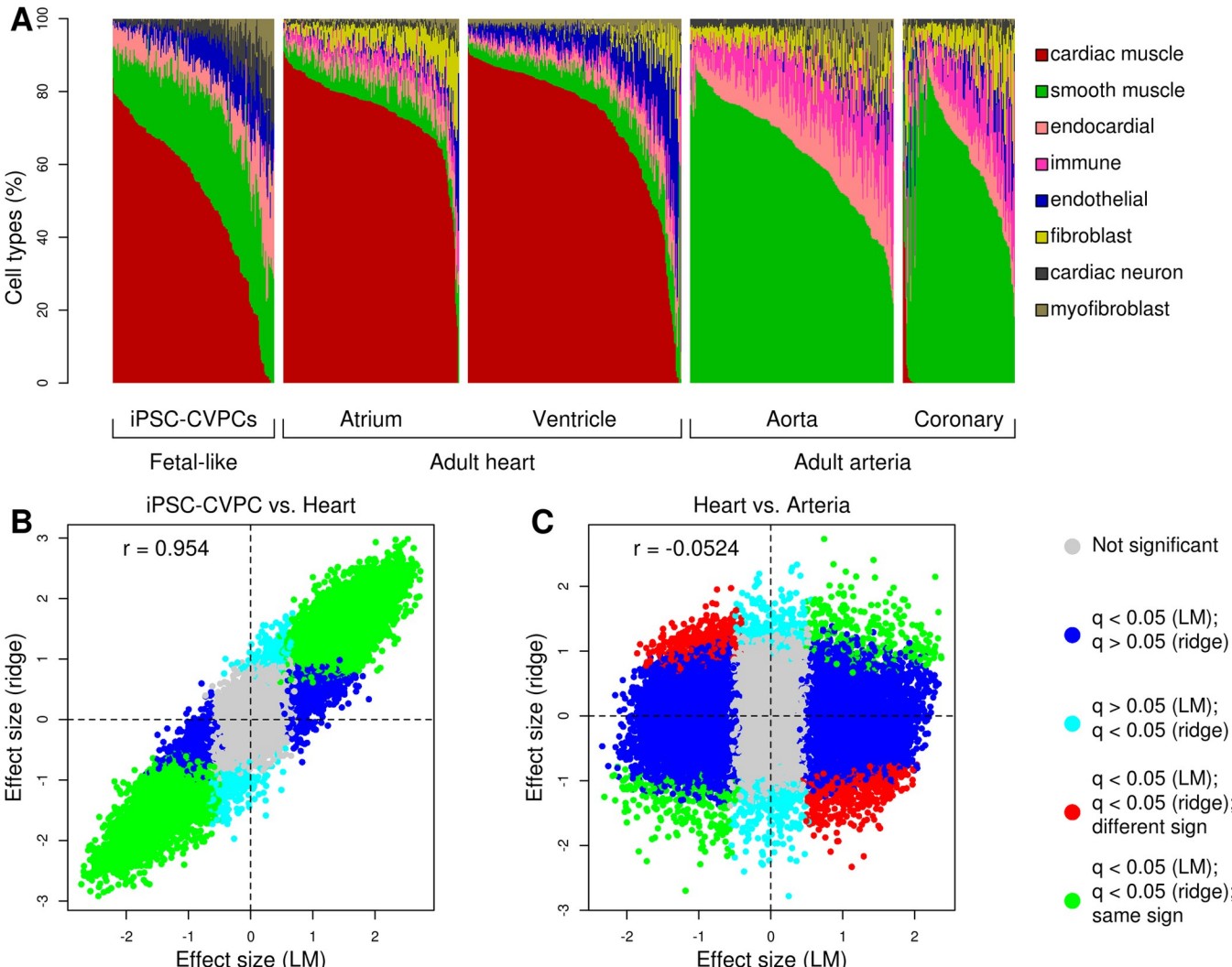

**Fig 4. Developmental stage-associated expression differences between fetal-like and adult heart.** (A) Estimated proportions of cell types across the 966 bulk RNA-seq samples. (B-C) Scatterplots showing the differential gene expression effect size considering cell type proportions as covariates (ridge regression, Y axis) versus without considering cell types (linear model, LM, X axis) for (B) iPSC-CVPC versus adult heart and (C) adult heart versus adult arteria. Genes differentially expressed in both analyses and have the same direction of effect are shown in green while genes with opposite direction of effect are in red; genes differentially expressed only using the LM are shown in blue; and genes that are differentially expressed only using ridge regression are in cyan.

showed that the most significantly enriched gene sets corresponded to the main function associated with each cell type (S14 Table). These results show that the 966 deconvoluted bulk RNA-seq samples have the expected cell-type proportions [14,52] and functional associations.

## Developmental stage-associated expression differences between fetal-like and adult heart

To determine the extent that the transcriptome expression differences between fetal-like iPSC-CVPC and adult heart (Figs 1C and S5A and S3 Table) were influenced by different cellular compositions, we performed the differential expression analysis using cell type proportions as covariates. Since the cell type proportions were not independent (i.e. each cell type is a linear combination of all the others), we performed this analysis using ridge regression, which

identified 12,781 genes, including 93 RBPs, and 12,680 isoforms as differentially expressed (FDR < 0.05, S15 Table). For both genes and isoforms there was a strong correlation between the effect size of the linear regression (which did not take cell type proportions into account) and the effect size of the ridge regression (r = 0.954, Figs 4B and S14), indicating that cell type composition only has a minimal effect on the observed developmental stage-associated expression differences between fetal-like iPSC-CVPC and adult heart. Conversely, between adult heart and adult arteria the ridge regression effect sizes were negatively correlated with the effect sizes of linear regression (r = -0.0524, Figs 4C and S14) indicating cellular composition is a major factor in gene and isoform expression differences between these two tissues. These analyses show that the observed developmental-stage-associated expression differences between fetal-like iPSC-CVPC and adult are only minimally affected by cellular composition differences, whereas the expression differences between the two adult CVS tissues are largely due to different cell type proportions.

## Differential RBP expression associated with developmental stage- and cell type-specific usage of isoforms

We next examined the extent that the fetal-like and adult heart developmental stage-associated genes and isoforms were also associated with specific cell-types (i.e., the same cell type within these two CVS tissues expressing different genes and isoforms). Ridge regression identified 3,727 genes (29.2% of the 12,781 differentially expressed genes, range 109–1,692 per cell type), 16 RBPs (17.2%, 1–14) and 454 isoforms (3.58%, 4–237), were associated with at least one of the six cell types present in both CVS stages (Fig 4A and S15 Table). Of note, this is likely an underestimate of the true number of cell-type- and development-stage-associated genes and isoforms. We further investigated developmental stage-specific gene and isoform expression in cardiac muscle and observed 998 genes (7.81% of the 12,781 differentially expressed genes; including 14 RBPs) and 176 isoforms (1.38%) that were associated with both CVS tissue (iPSC-CVPC versus adult heart) and cardiac muscle proportion (Fig 5A and 5B). Seven of the RBPs were positively associated with cardiac muscle proportion, of which five were overexpressed in fetal-like iPSC-CVPC (*MATR3*, *FXR1*, *RALYL*, *QKI*, *RMB24*) and two were overexpressed in adult heart (*RBFOX1*, *SAMD4A*, S16 Fig). The remaining seven RBPs were negatively associated with cardiac muscle proportion (i.e. overexpressed in a different cell type), of which five were overexpressed in fetal-like iPSC-CVPC (*FUS*, *PTBP3*, *RBM22*, *RBM4*, *SRSF7*) and two were overexpressed in adult heart (*PTBP1*, *SRSF9*). Known functions of several RBPs that were positively associated with cardiac muscle proportion are consistent with these observations: 1) *MATR3* is strongly expressed in the mouse heart, limb and brain during embryonic development, but is not expressed in their adult counterparts, and its disruption is associated with multiple congenital heart defects [53]; 2) *QKI* is involved in the regulation of apoptosis in cardiac muscle cells [54] and regulates cardiac myofibrillogenesis [11]; 3) *RBM24* regulates muscle-specific alternative splicing of genes involved in the sarcomere assembly and integrity during cardiac development, including *ACTN2*, *TTN* and *MYH10* [55,56]; and 4) *SAMD4A* plays important functions in both embryonic development and muscle homeostasis [57,58]. The two RBPs that were both negatively associated with cardiac muscle proportion and overexpressed in the adult heart (*SRSF9*, *PTBP1*) also had known functions consistent with these observations. *SRSF9* downregulation is known to promote muscle differentiation *in vitro* [59]. We examined the association between PTBP1 and its known target gene *EXOC7* [60] identifying two of *EXOC7* isoforms which differ only by the inclusion/exclusion of exon 7 (Fig 5C). The isoform expressing exon 7 (ENST00000589210.6_3) was overexpressed in iPSC-CVPC (p = 3.7 x $10^{-50}$) and positively associated with cardiac muscle proportion (p = 2.1

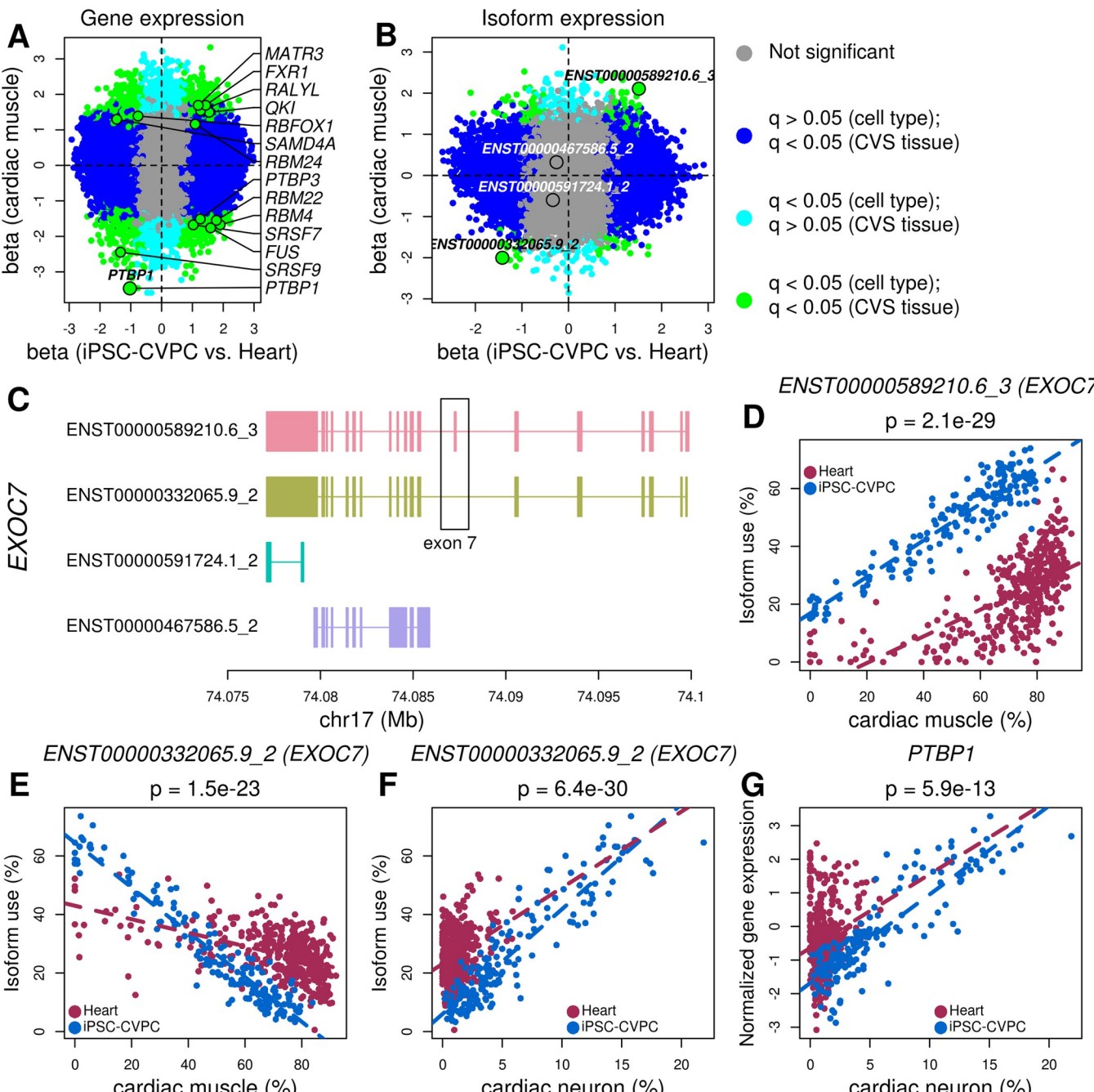

**Fig 5. Differential RBP expression associated with developmental stage- and cell type-specific usage of isoforms.** (A, B) Scatterplots showing differential (A) gene and (B) isoform expression effect size between iPSC-CVPC and adult heart (X axis) and the effect size of the association between expression and proportion of cardiac muscle (Y axis). Genes and isoforms significantly associated with both CVS tissue and cardiac muscle proportion are shown in green; genes and isoforms associated only with cell type are in blue; genes and isoforms that are associated only with CVS tissue are in cyan; genes and isoforms not associated with either CVS tissue or cell type proportion are shown in gray. *PTBP1* is shown in (A) and the four *EXOC7* isoforms are shown in (B). (C) Exon structure of the isoforms of *EXOC7*. The two isoforms (ENST00000589210.6_3 and ENST00000332065.9_2) differentially expressed between iPSC-CVPC and adult heart differ in the presence or absence of exon 7. (D-F) Scatterplots showing the association between isoform use (%) and cell type proportion for (D, E) cardiac muscle or (F) cardiac neuron (X axis) in iPSC-CVPC and adult heart (Y axis) for the two most commonly expressed isoforms of *EXOC7*. Dashed lines represent regression lines calculated on each CVS tissue. P-values represent the association between the use of each isoform and the cell type proportions (cardiac muscle in panels D and E, cardiac neuron in panel F). (G) Scatterplot showing the association between the normalized expression of *PTBP1* and the proportion of cardiac neuron (X axis) in iPSC-CVPC and adult heart. Dashed lines represent regression lines calculated on each CVS tissue. P-value represent the association between the *PTBP1* expression and the cardiac neuron proportion.

x $10^{-29}$), whereas the isoform that does not express exon 7 (ENST00000332065.9_2) was over-expressed in the adult heart (p = 3.7 x $10^{-40}$), negatively associated with the cardiac muscle proportion (p = 1.5 x $10^{-23}$) and positively associated with cardiac neuron (p = 6.4 x $10^{-30}$, Fig 5B, 5E and 5F and S15 Table). PTBP1 has been shown to inhibit *EXOC7* exon 7 inclusion most likely by binding to the acceptor site, which leads to exon skipping [60]. Here, we show that *PTBP1* has the same positive association (p = 5.9 x $10^{-13}$) with adult cardiac neuron as the *EXOC7* isoform that does not express exon 7 (ENST00000332065.9_2, Fig 5G), and is also similarly expressed at higher levels in the adult heart compared with iPSC-CVPC (p = 3.8 x $10^{-18}$). Overall, these analyses suggest that RBP developmental stage- and cell type-specific expression is associated with developmental stage- and cell type-specific usage of isoforms.

## Heart failure: Reversion to fetal-like RBP and isoform transcriptome

To determine if the CVS transcriptome during heart failure reverts to a fetal-like program [61,62] and the potential role of RBPs in this process, we investigated if RBP expression differences between heart failure samples pre- and post- implantation of mechanical support with a left ventricular assist device (LVAD, S16 Table) [20] resemble those between fetal-like iPSC-CVPC and adult heart. We first deconvoluted the cell types of 15 paired pre- and post-LVAD samples using CIBERSORT [51] and found that their cellular composition was overall comparable to the healthy adult GTEx heart samples (Fig 6A), except for smooth muscle (11.6% in post-LVAD samples compared with 6.8% in healthy adult heart, p = 4.2 x $10^{-4}$, S17 Table). We then integrated the fetal-like iPSC-CVPC and GTEx adult heart samples with the 15 paired pre- and post-LVAD samples and performed a principal component analysis using the expression levels of the 122 RBPs with known binding motifs [31–33]. We observed that the pre-LVAD heart failure samples were significantly closer to fetal-like iPSC-CVPC than post-LVAD (p = 1.3 x $10^{-6}$, paired t-test), whereas post-LVAD samples were significantly closer to the healthy adult heart samples (p = 2.7 x $10^{-8}$, paired t-test, Fig 6B and 6C).

To further characterize the role of RBPs in heart failure and if their expression reverts to a fetal-like state (Fig 1D–1J), we performed a differential expression analysis between the 15 pre-LVAD heart failure samples, fetal-like iPSC-CVPC and healthy adult heart using ridge regression taking cell type proportions into consideration (S18 Table). While four RBPs were overexpressed and nine were downregulated in heart failure compared with iPSC-CVPC, the mean effect size across all 122 RBPs was not different from zero (p = 0.273, t-test, Fig 6D and 6E) and hence, overall RBP expression was similar between the two tissues. Conversely, RBPs were expressed at higher levels in heart failure compared with healthy adult heart (p = 0.00128, t-test, Fig 6D and 6F) in a manner similar to their overexpression in iPSC-CVPC compared with heathy adult heart (p = 2.02 x $10^{-13}$, t-test, Fig 6D and 6G). All 14 RBPs differentially expressed between heart failure and healthy adult heart were also significantly differentially expressed between iPSC-CVPC and healthy adult heart, although three had opposite directions of effect (*HNRNPL*, *MBNL1*, *SAMD4A*). Five of the 14 RBPs differentially expressed between heart failure and healthy adult heart showed associations with both developmental-stage (fetal-like iPSC-CVPC versus adult heart) and cell-type, including three positively associated with cardiac muscle proportion and overexpressed in fetal-like iPSC-CVPC (*MATR3*, *QKI*, *RBM24*), one positively associated with cardiac muscle proportion and overexpressed in adult heart (*SAMD4A*, opposite direction of effect in heart failure), and one negatively associated with cardiac muscle proportion and overexpressed in healthy adult heart (*SRSF9*, Fig 5A and S15 Table). *QKI*, a gene involved in the inhibition of ischemia/reperfusion-induced apoptosis in cardiac muscle cells and involved in the regulation of the transition of fibroblasts toward profibrotic myofibroblasts in cardiac disease [54,63], was the most overexpressed RBP in both

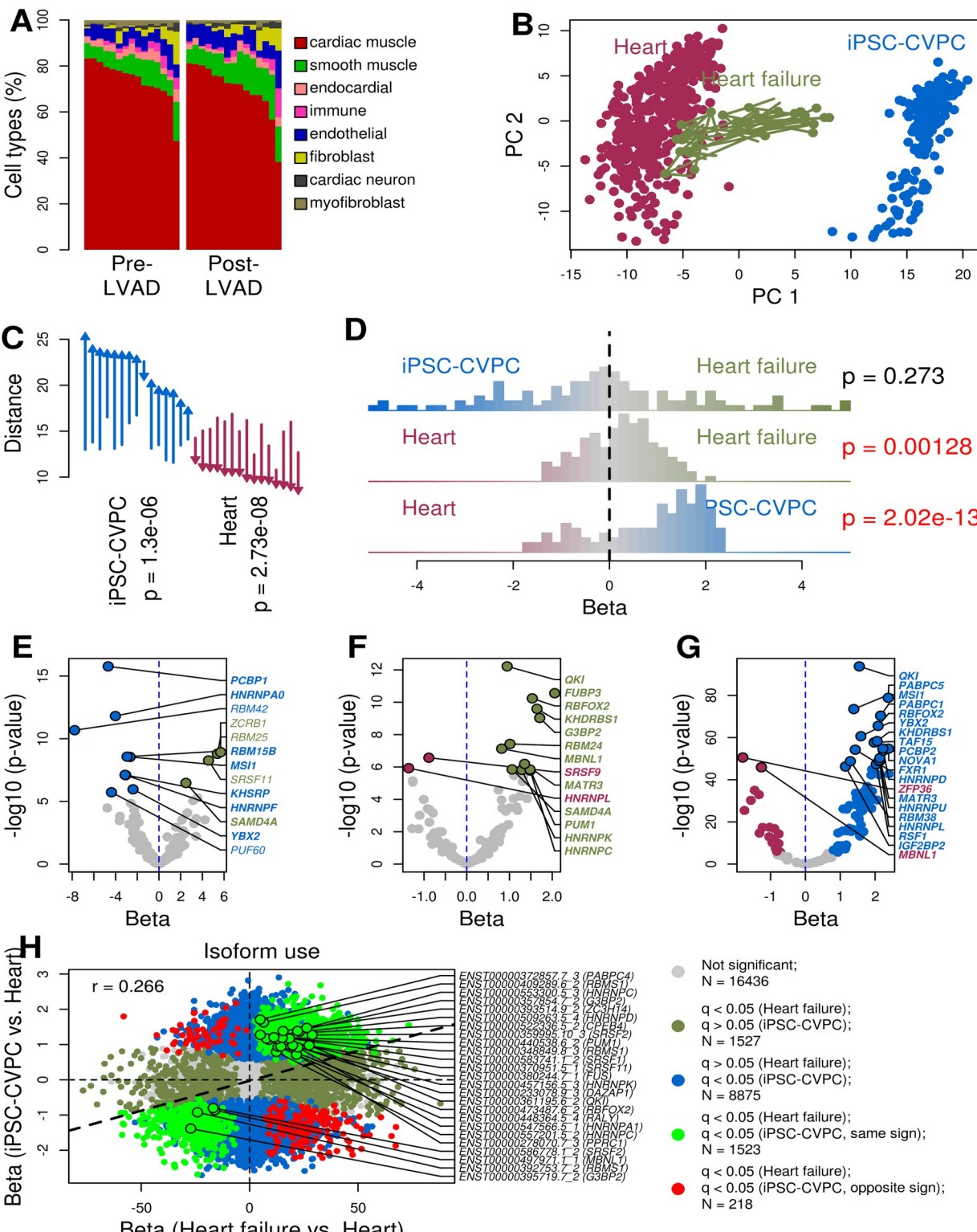

**Fig 6. Reactivation of fetal-specific RBPs and isoforms in heart failure.** (A) Barplot showing the cell type proportions in the 15 heart failure samples pre-LVAD (left) and post-LVAD (right). (B) Scatterplot showing the top two principal components (PC1: 46.1% variance explained; PC2: 20.5% variance explained) calculated on the expression of 122 RBPs in the integrated analysis of iPSC-CVPC, adult heart and heart failure samples. Arrows show the transcriptome changes from pre- to post-LVAD for all heart failure samples. (C) Differences in the average distance between iPSC-CVPC, adult heart and each of the 15 pairs of pre- and post-LVAD heart failure samples calculated on the expression of 122 RBPs

shown in B. Arrows indicate average distance and direction (e.g. arrows point up if distance increases) from pre-LVAD to post-LVAD to indicated tissue type. Distance was calculated on the top 10 principal components (87.2% total variance explained). P-values were calculated using paired t-test. (D) Histograms showing the differential expression (effect size) of the 122 RBPs calculated between each pair of CVS tissues (iPSC-CVPC, healthy adult heart and pre-LVAD heart failure). P-values were calculated using one-sample t-test in R (*t.test*, with parameter *mu = 0*). (E-G) Volcano plots showing differentially expressed RBPs (FDR < 0.05; effect size on the X axis; -log$_{10}$ p-value on the Y axis) between (E) pre-LVAD heart failure and iPSC-CVPC; (F) pre-LVAD heart failure and healthy adult heart; and (G) iPSC-CVPC and adult heart. In all three panels, the RBPs are color-coded for overexpression in iPSC-CVPC (blue), heart failure (green) and healthy adult heart (maroon). RBPs that are differentially expressed between iPSC-CVPC and healthy adult heart are shown in bold. In panel G, only the 20 RBPs with the most significant p-values are shown. (H) Scatterplot showing the correlation of the differential isoform usage expression between pre-LVAD heart failure and healthy adult heart (X axis) and between iPSC-CVPC and healthy adult heart (Y axis). On the right, shown are 25 isoforms for 20 different RBPs that were among those that reverted to the fetal-like iPSC-CVPC expression pattern in heart failure.

fetal-like iPSC-CVPC and heart failure, compared with healthy adult heart (p = 1.7 x 10$^{-94}$ and p = 6.1 x 10$^{-13}$, respectively, S18 Table). *RBM24*, which has been associated with dilated cardiomyopathy, one common cause of heart failure [64,65], was also overexpressed in heart failure. These data suggest that the differential expression of RBPs between heart failure and healthy adult heart is due to reactivation of fetal-specific RBPs (of which 35% display cell-type-associations), rather than a shift in cell-type composition.

Since we observed a strong overlap of RBP activity in fetal-like iPSC-CVPC and heart failure samples, we reasoned that the transcriptome-wide isoform switching that occurs between iPSC-CVPC and healthy adult heart (Fig 2A) may also occur between healthy heart and heart failure. To test this hypothesis, we performed a differential isoform expression analysis between heart failure, healthy adult heart and iPSC-CVPC. We observed 3,268 differentially expressed isoforms between heart failure samples and adult healthy heart (S18 Table), of which 1,523 (46.6%) were differentially expressed with the same trend as when comparing isoform usage between iPSC-CVPC and healthy adult heart, a number 1.85 times higher than expected considering that 10,616 isoforms (37.1% of all tested isoforms) were differentially expressed between iPSC-CVPC and healthy adult heart (odds-ratio = 1.85, p = 1.6 x 10$^{-56}$, Fisher's exact test, Fig 6H). Interestingly, 25 isoforms for 20 different RBPs were among those that reverted to the fetal-like iPSC-CVPC expression pattern in heart failure (S18 Table). Furthermore, the effect sizes of the differential isoform usage between heart failure and healthy adult heart across all isoforms were strongly correlated with the effect sizes of the differential isoform usage between fetal-like iPSC-CVPC and healthy adult heart (r = 0.266, p $\approx$ 0), indicating that in heart failure a transcriptome-wide isoform switch occurs, which results in the expression of thousands of fetal-specific isoforms.

## Discussion

To characterize the transcriptomic changes that occur in heart failure and their relationships with the fetal heart transcriptome, we initially investigated the gene expression and isoform differences between fetal-like iPSC-CVPC and adult CVS tissues (heart and arteria) and observed that thousands of genes and isoforms are differentially expressed. To validate our observations, we confirmed associations that have been previously reported as small-scale studies, including: 1) major cardiac development markers known to undergo isoform switching between fetal and adult heart (*SCN5A*, *TNNT2*, *ABLIM1* and *TTN*) express the appropriate isoforms respectively in fetal-like iPSC-CVPC and adult CVS tissues; 2) RBPs known to be involved in fetal cardiac development (including *CELF1*, *HNRNPL*, *RBFOX2* and *RBM24*) are overexpressed in fetal-like iPSC-CVPCs; 3) RBPs with known adult functions (*MBNL1* and *RBFOX1*) are overexpressed in adult CVS tissues; and 4) we confirmed the spatiotemporal context of the association between an RBP (*PTBP1*) and the alternative splicing of its known target gene (*EXOC7*) [60]. Here, using the expression levels of marker genes for eight cell types

obtained from single cell transcriptomic data, we were able to deconvolute cell type proportions in bulk RNA-seq samples for fetal-like iPSC-CVPC, healthy adult heart and arteria [15,51]. By taking cell type proportions into account, we were able to show that developmental-stage-associated expression differences between fetal-like iPSC-CVPC and adult CVS tissues are only minimally affected by cellular composition differences, whereas the expression differences between the two adult CVS tissues are largely due to different cell type proportions.

We show that heart failure samples have cell type proportions highly similar to healthy adult heart, with the main difference being a slightly higher fraction of smooth muscle cells in post-LVAD heart failure samples; however, we cannot exclude that this cellular difference may be due to sample collection, rather than reflecting a biological signal [20]. We compared the transcriptomes of heart failure samples with both fetal-like iPSC-CVPC and healthy adult heart and observed that heart failure samples have several characteristics that resemble the fetal heart, including the expression of cell-type- and developmental stage-specific RBPs and the usage of 1,523 fetal-specific isoforms. Our study indicates that transcriptome-wide isoform switching occurs between fetal and adult heart, and in heart failure a partial reversion to the fetal-specific isoform expression program occurs; and provides a valuable resource for the community to investigate isoform switching of individual genes between the fetal and adult healthy heart and during heart failure.

RBPs are a major driver of the underlying transcriptional changes between fetal, healthy and diseased adult heart. The significantly greater number of RBPs expressed in iPSC-CVPCs results in widespread isoform changes between the fetal-like and adult CVS tissues. While previous studies have identified only a relatively small number of genes that undergo isoform switching between fetal and adult heart [2,4,8,23–26,66–69], we show that thousands of isoforms are fetal- or adult-specific, indicating that alternative splicing plays an important role in driving the transition from fetal to post-natal and adult heart. Interestingly, fetal- and adult-specific isoforms have distinct characteristics, with fetal-specific isoforms being more likely to encode for known protein domains, to harbor RBP binding motifs, have canonical splice sites and contain typical upstream polypyrimidine tracts. These latter properties may be counterintuitive when combined with the overall higher RBP expression in iPSC-CVPC, because it would be expected that exons with stronger splice sites do not require the binding of RBPs for their inclusion. However, it has been shown that RBPs also play a role in post-transcriptional regulation and RNA transcript stability [31,70–73]. Overall, these observations suggest that early cardiac fetal development requires more strict transcriptional regulation than the adult.

Our study shows that in heart failure, reactivation of fetal-specific RBPs results in the expression of 1,523 fetal-specific isoforms which likely contributes to disease pathophysiology including reversion to a fetal-like metabolic state and oxygen consumption [61,62,74]. Therefore, RBPs in general, and in particular the 20 RBPs that have isoforms differentially expressed in both fetal-like iPSC-CVPC and heart failure compared with the healthy adult heart, may have two distinct uses in regenerative medicine: 1) to improve the quality and clinical relevance of iPSC-derived cardiac muscle cells; and 2) as novel therapeutic targets for heart failure. Since iPSC-CVPC are fetal-like [14,16,17], downregulating the expression of fetal-specific cardiac muscle RBPs could result in more mature adult-like cardiac muscle cells, thereby improving the use of iPSC-CVPC as a model system to study the adult heart *in vitro*. Experiments in primates have shown that transplantation of stem cell-derived cardiac cells into infarcted hearts improves cardiac regeneration but has several side-effects, including arrhythmia and ventricular tachycardia [75,76]; and clinical trials using stem cell-derived cardiac cells or their derivatives are currently underway [77–79]. Obtaining mature iPSC-CVPC for transplantation could improve their success in regenerative cardiac medicine. Furthermore, by pharmacologically targeting fetal-specific cardiac muscle RBPs or their isoforms in heart failure, it may be

possible to selectively revert diseased cardiac muscle cells to express a healthy adult transcriptome program. Although drugs specifically targeting cardiac muscle RBP isoforms have not yet been developed, several small molecules and specific antisense oligonucleotides have been shown to selectively inhibit RBPs aberrantly expressed in cancer, resulting in reduced cell viability, migration and/or invasion [80–84]. These promising studies suggest that future heart failure therapies may target fetal-specific isoforms of cardiac muscle RBPs to either mature and improve iPSC-CVPC for transplantation or directly revert transcriptomes in diseased cardiac muscle cells into healthy adult expression profiles.

## Methods

### RNA-seq data

We obtained RNA-seq data from two sources:

1. iPSC-CVPC were generated as part of the iPSCORE collection [14,85,86]. All transcriptomic, epigenomic and genetic variation data generated within the iPSCORE collection has been deposited in dbGaP (phs000924 and phs001325).

2. Adult CVS samples (aorta, coronary arteries, atrial appendages and left ventricles) were obtained from GTEx (dbGaP: phs000424).

**iPSCORE iPSC-CVPC samples.** The 139 iPSCORE subjects [86] included members of 27 families (2–9 members/family), 7 monozygotic twin pairs, and 55 singletons (S1 Table). During recruitment of iPSCORE subjects, subject information (sex, age, family, ethnicity, and cardiac diseases) was collected. Recruitment was approved by the Institutional Review Boards of the University of California, San Diego and The Salk Institute (project no. 110776ZF). Skin biopsies were collected for each iPSCORE individual.

As previously described in detail [86–88], we reprogrammed skin fibroblast samples from 139 iPSCORE individuals using non-integrative Cytotune Sendai virus (Life Technologies) and the 149 iPSCs (7 subjects had 2 or more clones each) were shown to be pluripotent and to have high genomic integrity with no or low numbers of somatic copy-number variants (CNVs). To generate iPSC-derived cardiovascular progenitors (iPSC-CVPC) we used a small molecule differentiation protocol, which included purification of iPSC-CVPC cultures using lactate [89], and harvested iPSC-CVPC at day 25 (D25) differentiation [14,85]. We obtained RNA-seq (150 bp PE) for 180 iPSC-CVPC (30 of the 149 iPSCs were differentiated two or more times). All iPSCORE iPSC-CVPC RNA-seq samples are available through dbGaP (phs000924).

**GTEx adult CVS samples.** FASTQ files for 786 GTEx RNA-seq (75 bp PE) datasets were downloaded from dpGaP (phs000424). GTEx samples were obtained from postmortem tissues from 352 individuals and included 227 aorta, 125 coronary artery, 196 atrial appendage and 238 left ventricle (S1 Table).

**Data processing.** For gene expression, we obtained transcript per million bp (TPM) as previously described [14,90]. Briefly, FASTQ files were aligned to the hg19 reference genome using STAR 2.5.0a [91] and Gencode V.34lift37 [92] with parameters *outFilterMultimapNmax 20,–outFilterMismatchNmax 999,–alignIntronMin 20,–alignIntronMax 1000000,–alignMatesGapMax 1000000*. We sorted the BAM files using Sambamba 0.6.7 [93] and marked duplicates using biobambam2 (2.0.95) *bammarkduplicates* [94]. To quantify gene expression and isoform abundance, we used RSEM [95] with options *rsem-calculate-expression—bam—num-threads 16—no-bam-output—seed 3272015—estimate-rspd—paired-end—forward-prob 0*. Using this pipeline, we determined the expression of 62,492 genes and their corresponding 229,835 isoforms.

We tested if the read length differences between the iPSC-CVPC and adult CVS tissues (150 and 75 bp paired-end, respectively) could affect estimated gene and isoform expression levels. We randomly selected 50 iPSCORE RNA-seq samples and trimmed the 3' end of their read length to 75 bp (S2 Table). We determined gene and isoform expression levels in these 50 samples using the pipeline described above. Next, we tested the expression differences between these 75 bp-read samples and their corresponding 150 bp-read samples using paired t-test with Bonferroni correction for FDR. We identified 299 genes and 8,999 isoforms (corresponding to 949 genes) with significant expression level differences (1,059 genes in total, S19 Table). We could not use read length as a covariate differential expression analysis, as the biological differences between iPSC-CVPC and adult hearts would have been nullified due to perfect correlation between the "read length" variable and "developmental stage". We therefore removed the genes and isoforms whose expression levels were associated with read length.

**Integrated analysis of iPSCORE and GTEx RNA-seq datasets.** We generated one matrix that integrated the expression levels of genes and isoforms in each of the 966 CVS RNA-seq samples (180 iPSCORE and 786 GTEx). To remove genes expressed at low levels and/or only in a small subset of samples, we only considered genes "expressed" if they had a TPM $\geq 1$ in at least 10% of samples. For isoform analysis, only expressed genes with at least two expressed isoforms (usage >10% in at least 10% of samples) were used. We used isoform usage %, rather than isoform TPM, because isoform use is independent of the transcript length and therefore more comparable across samples [95]. This resulted in a total of 20,393 genes and 38,271 isoforms (corresponding to 10,686 genes) used in the differential expression analysis.

We quantile-normalized gene and isoform expression levels using the *normalize.quantiles* (preprocessCore) and *qnorm* functions in R, in order to obtain mean expression = 0 and standard deviation = 1 for each gene and isoform.

All the analyses described in this study were performed on the gene expression and isoform use data integrated between the iPSCORE and GTEx collections.

Matrices including all TPM, quantile-normalized gene expression, isoform usage and quantile-normalized isoform usage can be found at https://doi.org/10.6084/m9.figshare.13537343.

## Transcriptome-wide gene expression and isoform usage differences between CVS tissues

We used multiple approaches to determine the transcriptomic differences between fetal-like iPSC-CVPC and adult CVS tissues. Specifically:

1. To determine the transcriptome-wide differences between iPSC-CVPC and adult CVS tissues, we performed dimensionality reduction, clustering, cell cycle scoring and pseudotime estimation;

2. We calculated pairwise differential gene expression between the three CVS tissues (iPSC-CVPC, adult heart and adult arteria) using linear regression;

3. We determined pairwise differential isoform usage between the three CVS tissues (iPSC-CVPC, adult heart and adult arteria) using linear regression;

4. We performed functional enrichment analysis to validate the observed gene expression and isoform usage differences between developmental stages;

5. We tested the overlap between exons specific for each of the three CVS tissues and protein domains to investigate functional differences between iPSC-CVPC-specific and adult CVS-specific isoforms.

**Dimensionality reduction of iPSCORE and GTEx samples.** We performed dimensionality reduction to examine global transcriptome-wide gene expression differences between the 966 bulk RNA-seq samples (S1 Fig and S1 and S2 Tables). We conducted principal component analysis (PCA) of the 966 bulk RNA-seq samples using Seurat [28]. We created a Seurat object using TPM values as input and employed log-normalization and scaling using default parameters of Seurat. We identified the 2,000 most variable features from variance stabilizing transformation (VST) and performed dimensional reduction to obtain 50 principal components. To understand how samples cluster together, we implemented a graph-based approach to cluster samples at different resolutions of 0, 0.01, 0.2, and 0.7. *Clustree* [96] was used to build a clustering tree, and dimensional reduction was visualized using UMAP (Figs 1A and S2–S4). The top 50 PCs and UMAP coordinates were extracted from the Seurat object using the *Embeddings* function. Cardiomyocytes divide at high rates during fetal development and early childhood, then cell division decreases gradually to <1% per year in adulthood [27]. To examine cell division rates in iPSC-CVPC compared with the two adult tissues, we used the *CellCycleScoring* function in Seurat with the reference G2/M and S phase markers in the *cc.genes* object [28,29] and found that iPSC-CVPC were dividing at a significantly higher rate (G2M and S-scores respectively, heart: $p = 3.1 \times 10^{-83}$, $p = 1.2 \times 10^{-84}$, arteria: $p = 1.7 \times 10^{-79}$, $p = 1.6 \times 10^{-79}$, Mann-Whitney U test, S2F and S2G Fig and S2 Table).

**Estimation of pseudotime using Monocle.** To estimate the time reflected by the different developmental stages of the 966 CVS samples, we performed pseudotime analyses using Monocle [30]. Pseudotime was calculated based on the expression of all expressed genes following the standard workflow for constructing developmental trajectories. Pseudotime trajectory was ordered by rooting time (i.e. pseudotime = 0) in iPSC-CVPC using the *GM_state* function. We observed that iPSC-CVPC were significantly at an earlier developmental stage than the two adult CVS tissues (heart: $p = 7.3 \times 10^{-85}$, arteria: $p = 1.5 \times 10^{-79}$, Mann-Whitney U test, S2H Fig and S2 Table).

**Differential expression analysis between CVS tissue types.** The three CVS tissue types (iPSC-CVPC, adult heart, adult arteria) are composed of the same cell types but in different proportions, have significantly different ratios of male:female individuals ($p = 3.7 \times 10^{-7}$, Fisher's exact test, S1A Fig) and the same cell types may have different numbers of mitochondria at different developmental stages. For example, cardiac muscle cells have more mitochondria than other cell types found in cardiac tissues, which results in a larger number of RNA-seq reads mapping to the mitochondria and, in turn, a smaller number of reads mapping to autosomes and sex chromosomes. This would result in lower TPM values for samples with high mitochondrial content. To minimize the potential batch effects due to these biases, we used the following covariates: 1) sex; 2) normalized number of RNA-seq reads; 3) % of reads mapping to autosomes or sex chromosomes; and 4) % of mitochondrial reads. We performed differential expression analysis between each pair of tissues (iPSC-CVPC vs. adult heart; iPSC-CVPC vs. adult arteria; and adult heart vs. adult arteria) using a linear model (*lm* function in R) and considering these multiple covariates:

$$Y_i = \beta_j T_{ij} + \sum_{p=1}^{P} \gamma_p C_{ip} + \epsilon_{ij}$$

Where $Y_i$ is the phenotype value (normalized gene expression or % isoform use) for sample $i$, $\beta_j$ is the effect size tissue $j$, $T_{ij}$ is the tissue type of sample $i$, $C_{ip}$ is the value of the $p^{\text{th}}$ covariate for sample $i$, $\gamma_p$ is the effect size of the $p^{\text{th}}$ covariate, $P$ is the number of covariates used, and $\epsilon_{ij}$ is the error term for sample $i$ at tissue $j$. P-values were corrected using Bonferroni's method (S3 Table).

Differential isoform usage analysis (S5 Table) was performed using the same linear model described above.

Several small-scale studies described changes in gene expression of specific RBPs and isoform switching of specific cardiac genes between fetal and adult CVS tissues. As shown in Figs 1 and 2, our findings are in agreement with these previous studies and identify thousands of novel developmental stage-specific genes and isoforms.

**Functional enrichment analysis.** Functional enrichment analysis was performed on 12,249 gene sets obtained from MSigDB V.7.1 [97]. We used gene sets included in the following collections (collection IDs are indicated in parenthesis): Hallmark (h.all); Gene Ontology (GO) biological process (c5.bp), cellular component (c5.cc) and molecular function (c5.mf); canonical pathways included in Biocarta (c2.cp.biocarta), KEGG (c2.cp.kegg) and Reactome (c2.cp.reactome). We performed gene set enrichment analysis (GSEA): for each gene set, we performed a t-test between the effect size ($\beta_j$ in the formulas described above) of each gene included in the gene set and all the expressed genes not included in the gene set. P-values were corrected using Benjamini-Hochberg's method. We determined the functional enrichment for tissue-associated genes (iPSC-CVPC vs. adult heart; iPSC-CVPC vs. adult arteria; and adult heart vs. adult arteria) and cell type-associated genes.

To determine the functional enrichment of genes with CVS developmental stage-specific isoforms, we obtained all the genes with one or more differentially expressed isoforms (FDR < 0.05 and $|\log_2$ ratio$| > 2$) and all the genes that did not have any differentially expressed isoform (FDR > 0.05) and compared their associations with each gene set using Fisher's exact test. P-values were corrected using Benjamini-Hochberg's method (S4 and S6 Tables).

**Identifying CVS developmental stage-specific exons.** To identify exons expressed only in iPSC-CVPC compared with adult heart, we selected all genes that had at least one isoform that was iPSC-CVPC-specific (FDR < 0.05 and $\log_2$ ratio > 2 in S5 Table) and at least one isoform that was adult heart-specific (FDR < 0.05 and $\log_2$ ratio < -2 in S5 Table). Gencode provides unique identifiers for each exon, with exons having the same ID if they belong to different isoforms but have the same coordinates. Therefore, we intersected the lists of exons from the iPSC-CVPC-specific and adult heart-specific isoforms and considered as iPSC-CVPC-specific all the exons that were not present in the adult heart-specific isoforms. Likewise, we defined as adult heart-specific exons all the exons that were present only in the adult heart-specific isoform but not in the iPSC-CVPC-specific. We performed similar filtering to obtain CVS tissue-specific exons between iPSC-CVPC and adult arteria and between adult heart and adult arteria.

**Finding overlap between CVS developmental stage-specific exons and protein domains.** We obtained the mapping on genomic coordinates for >700,000 protein domains from the Prot2HG database [44] and intersected their coordinates with the coordinates of the CVS tissue-specific exons from each pairwise analysis (iPSC-CVPC vs. adult heart, iPSC-CVPC vs. adult arteria and adult heart vs. adult arteria, S7 Table). To determine enrichment in the fraction of exons that encode for known protein domains between each pair of CVS tissues, we performed Fisher's exact test between each pair of tissues. While we did not observe enrichments for specific protein domains, iPSC-CVPC-specific exons were in significantly more likely to overlap functional protein domains than their adult-specific counterparts.

## RBP and alternative splicing analyses

Our functional enrichment analysis identified RBPs as the gene set most significantly differentially expressed between iPSC-CVPC and both adult heart and adult arteria. To

characterize the functional differences of RBPs between fetal-like and adult CVS tissues, we determined:

1. The associations between isoform tissue-specific expression and the occurrence of binding sites for 38 RBPs (measured by enhanced CLIP);

2. The enrichment, measured using Homer *findMotifsGenome.pl*, near splice sites of fetal-specific exons of motifs for 122 RBPs in the RNA binding gene set (GO:0003723);

3. The differences in the splice donor and acceptor site sequences for fetal- and adult-specific exons.

**Enrichment of CVS developmental stage-specific isoforms for RBP binding sites.** We obtained *in vitro* binding activity of 39 RBPs using enhanced CLIP (eCLIP, 59 experiments) from a recent ENCODE study [31]. One RBP (*MATR3*) was not expressed in CVS tissues and was removed from further analyses. We downloaded BED files corresponding to biologically reproducible eCLIP peaks and intersected them with the genomic coordinates of the loci encoding each gene (defined as the start-end coordinates obtained from Gencode) expressed in the 966 cardiac samples. To test whether a CVS tissue was enriched for having active RBP binding sites, we performed pairwise comparisons (iPSC-CVPC vs. adult heart, iPSC-CVPC vs. adult arteria and as a control adult heart vs. adult arteria) using the following approach:

1. For each pairwise comparison, we obtained the list of isoforms overexpressed in tissue 1 and those overexpressed in tissue 2, and converted them to their associated genomic coordinates. To examine overexpressed isoforms, we used 13 filtering thresholds, from $\log_2$ ratio = 0 (the least stringent, including all the isoforms that had a significant p-value after FDR correction in S5 Table) to $\log_2$ ratio > 6 (Figs 3B and S8), in 0.5 increments.

2. As background, we used all the genes that did not have any differentially expressed isoforms in the pairwise comparison between tissue 1 and tissue 2.

3. We intersected the three lists of genomic coordinates with the coordinates of eCLIP peaks for each eCLIP experiment.

4. We compared the fraction of genes associated with tissue 1 against background using Fisher's exact test (*fisher.test* in R); likewise, we compared the fraction of genes associated with tissue 2 against background.

**RBPs with known motifs.** We obtained the position weight matrix (PWM) of RBPs included the RNA binding gene set (GO:0003723) from three sources: 1) RBPmap V.1.1 [33]; 2) ENCODE [31]; and 3) CISBP-RNA V.0.6 [32]. For the analyses described in the manuscript, we considered only the 122 RBPs that both had a known motif (262 PWMs in total) obtained from these sources and were expressed in the 966 cardiac samples.

**Enrichment of CVS developmental stage-specific exons for RBP motifs.** To determine whether tissue-specific exons were enriched with RBP binding motifs, we created a 100-bp window upstream of exons that are specific to iPSC-CVPC, the adult heart or the adult arteria. The first exon of each isoform were excluded. We created a custom motif file, consisting of 262 RBP motifs (corresponding to 122 distinct RBPs) from RBPmap, ENCODE and CISBP-RNA. We tested enrichment using Homer *findMotifsGenome.pl* [98] with parameters -*mknown [custom RBP PWM file] -mcheck [custom RBP PWM file] -len 4,5,6,7,8,10,12* in a pairwise fashion (iPSC-CVPC vs. adult heart, iPSC-CVPC vs. adult arteria, and adult heart vs. adult arteria, S9 Fig). Enrichment was calculated as the $\log_2$ ratio between the proportion of exons specific for *tissue 1* and those specific for *tissue 2*. Since Homer *findMotifsGenome.pl* tests only for enrichment

(but not depletion) of *tissue 1* against background, to determine enrichment in each CVS tissue we performed two analyses and then integrated their results: 1) to test the enrichment for *tissue 1*, we used *tissue 2* as background; and 2) to test the enrichment for *tissue 2*, we used *tissue 1* as background. To integrate the two analyses, for each RBP motif, we retained only the analysis with the most significant q-value and adjusted the sign of the $\log_2$ ratios, with positive values representing enrichment for *tissue 1* and negative values enrichment for *tissue 2*.

**Finding differences in the splice donor and acceptor site sequences.** To determine the consensus sequences at the splice donor and acceptor sites of CVS tissue-specific exons, we obtained the coordinates of 100 bp upstream and downstream of each exon (see section above "Identifying CVS developmental stage-specific exons") and converted them to FASTA using *bedtools getfasta*, with options–*s*–*tab* to obtain strand-specific sequences in a tab-separated text file. We performed pairwise comparison between each pair of CVS tissues (iPSC-CVPC vs. adult heart, iPSC-CVPC vs. adult arteria and adult heart vs. adult arteria). We calculated the frequency of each of the four nucleotides at each position for exons specific for each of the two tested CVS tissues and performed Fisher's exact tests on each nucleotide to calculate enrichment. In Fig 3D–3I, we only show the enrichment values for the most common nucleotide at each position, while S10 Table shows all the tests.

## CVS tissue cellular deconvolution and validation of cell-type proportions

We estimated the cell type composition of each sample and performed linear regression using the proportion of each cell type as covariates.

**Cell composition estimation.** To determine the marker genes and their expression levels to use as input for cell type composition deconvolution, we used our recently published pipeline [15], which relies on scRNA-seq data from the *Tabula muris* collection [50]. Although several studies using scRNA-seq in human cardiac tissues have been published, these data are extremely sparse (<50,000 reads/cell) and hence generate less accurate deconvolution results compared with *Tabula muris* (>800,000 reads/cell). We first obtained a Seurat object containing normalized gene expression data from 4,365 FACS-sorted cardiac cells sequenced at high read depth in the *Tabula Muris* collection [50] (S10 Fig). We then used the *FindAllMarkers* function in Seurat with parameters *only.pos = TRUE, min.pct = 0.2, logfc.threshold = 0.1* to determine marker genes in each of the eight cell types defined by *Tabula Muris*: cardiac muscle, cardiac neuron, endocardial, endothelial, fibroblast, immune, myofibroblast and smooth muscle. We found 7,576 genes overexpressed in at least one cell type and combined their expression levels using the Seurat function *AverageExpression* to calculate their average expression in each cell type. To perform cell type deconvolution in all 966 CVS samples, we ran CIBERSORT [51] using the average single cell expression levels of all marker genes in each cell type as "signature gene matrix" and their TPM from bulk RNA-seq as "mixture matrix" (Fig 4A and S11 and S12 Tables). We ran CIBERSORT with quantile normalization = FALSE.

**Validation of the estimated cell type proportions.** To validate the estimated cell type proportions determined using CIBERSORT, we determined the associations between the expression of each gene and the proportion of each cell type, then we used these associations to: 1) test correlation with gene expression in two scRNA-seq datasets; and 2) perform GSEA. To determine the associations between gene expression and each cell type proportion, we used a linear model (*lm* function in R, S12 Fig and S13 Table):

$$Y_i = \beta_j E_{ij} + \sum_{p=1}^{P} \gamma_p C_{ip} + \epsilon_{ij}$$

Where $Y_i$ is the phenotype value for sample $i$, $\beta_j$ is the effect size cell type $j$, $E_{ij}$ is the proportion of cell type $j$ in sample $i$, $C_{ip}$ is the value of the $p^{th}$ covariate for sample $i$, $\gamma_p$ is the effect size of the $p^{th}$ covariate, $P$ is the number of covariates used, and $\epsilon_{ij}$ is the error term for sample $i$ at cell type $j$. We used the following covariates: 1) sex; 2) normalized number of RNA-seq reads; 3) % of reads mapping to autosomes or sex chromosome; 4) % of mitochondrial reads. P-values were corrected using Bonferroni's method.

To validate the cell type proportions, we used scRNA-seq analysis of 32,026 cells from eight iPSC-CVPC samples [14] and 33,050 cells from 11 adult left ventricle samples [52]. We used CellRanger to aggregate the eight iPSC-CVPC and the 11 adult ventricle samples (S13 Fig). Since these two datasets were obtained using different methods from different groups and had different read depths (iPSC-CVPC: 3,030 median UMI/cell and 1,333 expressed genes/cell; adult left ventricle: 463 median UMI/cell and 210 expressed genes/cell post-aggregation), we could not integrate them without confounding real biological differences with batch effects. Therefore, we analyzed them separately to validate the differential expression results that we obtained using bulk RNA-seq.

We aimed at assessing whether the associations between cell type proportion and gene expression could be confirmed using scRNA-seq. To do this, we measured the mean expression in each cell type associated with >100 cells (cardiomyocytes in iPSC-CVPC, cardiac muscle, fibroblasts, endothelium and smooth muscle in the adult heart) and calculated the $\log_2$ ratio against the mean expression in all the other cells. Mean expression was calculated as *mean(expm1(x)) + 1* in R, where *x* is a vector of expression values across all cells and *expm1* is a function that computes the exponential of a given value minus one. Next, we measured the correlation between the $\log_2$ ratio calculated in the scRNA-seq samples and the effect size of the association between stage and gene expression in bulk RNA-seq. This correlation was calculated 1) across all genes with at least one read assigned to them in the scRNA-seq; and 2) using only highly expressed genes. Since read depth was different between iPSC-CVPC and adult left ventricle, we used different thresholds: we considered as "highly expressed" all genes expressed in at least 50% of cells for iPSC-CVPC and 10% for adult left ventricle (S13 Fig). For all cell types, we observed a significantly positive correlation across all genes and all highly expressed genes, confirming that the associations between gene expression levels and estimated cell type proportions in bulk RNA-seq are in agreement with gene expression level differences between cell types in scRNA-seq.

To confirm the differences in the tissue-specific associations between cardiac muscle proportion and gene expression, we obtained the number of reads mapping to each gene in cardiomyocytes (CM) from iPSC-CVPC and adult cardiac muscle single cells. Next, we normalized the gene expression levels in the two datasets using the *calcNormFactors* function in *edgeR* [99] and computed $\log_2$ ratio between the two stages. We measured the correlation between these $\log_2$ ratio values and the difference between effect sizes of the association between cardiac muscle proportion and gene expression in bulk RNA-seq for iPSC-CVPC and the association between cardiac muscle proportion and gene expression in bulk RNA-seq for adult heart. The correlation was significantly positive (R = 0.202, p = 4.8 x $10^{-162}$, S13H Fig), indicating that the CVS developmental stage-specific associations between gene expression and cell type proportions reflect gene expression differences at single-cell resolution.

We performed GSEA using the method described above (see section "Functional enrichment analysis"). Briefly, for each gene set, we performed a t-test between the effect size ($\beta_j$ in the formulas described above) of each gene included in the gene set and all the expressed genes not included in the gene set. P-values were corrected using Benjamini-Hochberg's method. This analysis shows that each cell type is associated with genes with relevant functions (S14 Table).

## Developmental stage- and cell-type-associated gene expression and isoform usage

To examine whether the transcriptional differences between the fetal-like iPSC-CVPC and two adult CVS tissues were due to expression of developmental stage-specific genes and isoforms, while the transcriptional differences between the two adult CVS tissues were a consequence of their cellular composition differences, we determined the association between gene expression and isoform usage with CVS tissue, taking cell type proportions into consideration. We used ridge regression (*ridge.proj* from the *hdi* package in R), with the following covariates: sex; normalized number of RNA-seq reads; % of reads mapping to autosomes or sex chromosome and % of mitochondrial reads; and the cell type proportions of each of six cardiac cell types (S15 Table). We used ridge regression, rather than linear regression, because it does not require independence between covariates and each cell type proportion is a linear combination of all the others. P-values were FDR corrected with Bonferroni's method. Fibroblasts and immune cells were not present in iPSC-CVPCs, therefore it was not possible to distinguish associations between gene expression and cell-type proportions from associations between gene expression and developmental stage. These two cell types were therefore not considered in the analyses described in the section "Differential RBP expression drives CVS developmental stage- and cell type-specific usage of isoforms".

## Comparing heart failure with fetal-like and adult heart transcriptomes

To validate the transcriptional changes between adult and fetal gene programs in post-natal pathophysiological conditions, we obtained FASTQ files for bulk RNA-seq of 8 non-ischemic and 7 ischemic heart failing samples before and after mechanical support with left ventricular assist device (i.e. 15 pairs, 30 total samples; GEO ID: GSE46224) [56]. Reads (100 bp, paired end) were processed with the same pipeline described above for the 966 CVS samples, including the identification of expressed genes and isoforms. All samples had properly paired reads > 98% and percent mRNA reads > 98%. Sample identity check was performed with *plink genome* [100] using SNPs with read depth > 10. All 15 sets of paired samples were confirmed as being derived from the same individual (PI_HAT > 98%).

Next, we performed integrative principal component analysis (PCA) on the 30 heart failure samples and the healthy CVS samples (iPSC-CVPC and adult heart) samples using Seurat's Reference-based Integration Workflow [28] where healthy CVS samples were used as reference (Fig 6B and S16 Table). We performed integration using TPM values for 121 RBPs (one RBP, *ESRP2*, was not expressed in the heart failure samples). Integration was performed using the standard Seurat workflow (https://satijalab.org/seurat/v3.1/integration.html) on the expression of all 121 RBPs. Principal components were extracted from the integrated object using the Seurat *Embeddings* function.

We calculated pairwise distance between each pair of the 996 samples (966 healthy and 30 heart failure samples) using the *dist* function in R on the top 10 principal components (S16 Table). Next, we obtained the mean distance each of the 30 heart failure samples against iPSC-CVPC, adult heart or adult arteria samples (Fig 6C). To determine whether pre- and post-LVAD samples differed in their distance from each of the three cardiac stages, we performed paired t-test.

Differential expression analysis was performed using ridge regression, as described above (S18 Table).

## Supporting information

**S1 Fig. Description of the subjects included in this study.** (A) Sex distribution between iPSCORE individuals (blue) and GTEx individuals (maroon). (B) PCA showing the ancestry

of all subjects. Colored circles represent ancestry of individuals from the 1000 Genomes Project. GTEx and iPSCORE individuals are represented by pink and blue "X", respectively. Details are in S1 Table.
(TIFF)

**S2 Fig. Transcriptomic differences between fetal-like iPSC-CVPC and adult CVS tissues.**
(A-E) UMAP plots showing how the 966 bulk RNA-seq CVS samples cluster at varying resolutions (S3 Fig). Panel (A) shows the samples according to their tissue of origin (iPSC-CVPC, adult atrium, ventricle, aorta or coronary artery). In panels (B-E) samples are colored based on clustering (B) at low resolution (resolution = 0); (C) at intermediate-low (resolution = 0.01); (D) at intermediate-high resolution (resolution = 0.2); and (E) at high resolution (resolution = 0.7). Low resolution separated the samples based on their developmental stage (iPSC-CVPC and adult tissues); intermediate-low resolution divided the samples into three clusters corresponding to iPSC-CVPC, adult heart (atrial appendage and left ventricle samples) and adult arteria (aorta and coronary artery samples); intermediate-high resolution separated the samples into the five CVS tissues (iPSC-CVPC, adult atrium, ventricle, aorta or coronary artery); and high resolution separated samples derived from the same tissue-type suggesting the presence of cellular heterogeneity across samples. To increase statistical power, for all downstream analyses we used the clusters obtained at intermediate-low resolution (i.e., we combined atrium and ventricle samples into one group referred to as "adult heart", and aorta and coronary artery into one group referred to as "adult arteria"). There were 5 adult heart samples that cluster with the adult arteria and 1 adult arteria sample that clusters with the adult heart. Most likely, their cell type composition differs from the other samples of the same label due to the area in which they were collected. We retained the original labeling of the 5 samples as adult heart and the 1 sample as adult arteria for downstream analyses. (F-H) Boxplots showing (F) G2M score, (G) S phase score and (H) pseudotime calculated using Seurat or Monocle. iPSC-CVPC display a higher G2M score compare with both heart and arteria (heart: $p = 3.1 \times 10^{-83}$, arteria: $p = 1.7 \times 10^{-79}$, Mann-Whitney U test), a higher S score (heart: $p = 1.2 \times 10^{-84}$, arteria: $p = 1.6 \times 10^{-79}$, Mann-Whitney U test) and a lower pseudotime (heart: $p = 7.3 \times 10^{-85}$, arteria: $p = 1.5 \times 10^{-79}$, Mann-Whitney U test).
(TIFF)

**S3 Fig. Hierarchical clustering of 966 RNA-seq CVS samples at different resolutions.** Cluster trees showing how 966 RNA-seq samples are grouped together with KNN clustering at different resolutions. The nodes are color-coded in the same manner as the clusters in the companion UMAP plots in S2A–S2E Fig. Lower resolution separates the samples based on developmental stage, while intermediate and higher resolutions resolve iPSC-CVPC (right) and adult CVS (left) samples into sub-clusters.
(TIF)

**S4 Fig. Principal components capture transcriptomic differences between CVS tissues.**
Corrplot [101] showing the Pearson correlation between the top 5 principal components calculated on the 966 CVS bulk RNA-seq samples and binary vectors indicating the type of tissue —aorta, coronary, atrium, ventricle, heart (atrium and ventricle), arteria (aorta and coronary), and iPSC-CVPC (S2 Table). The legend indicates the strength of the correlation for each pairwise comparison with blue representing a positive correlation and red representing a negative correlation. This correlation analysis shows that that the top principal component (PC1, 32.2% variance explained) is strongly associated with CVS developmental stage, whereas PC2 (19.4% variance explained) divides the adult CVS samples between heart and arteria. These results confirm that the two adult heart tissues (atrium and ventricle), as well as the two adult arteria

tissues (aorta and coronary) have similar transcriptomes and supports combining samples in each of the three clusters obtained at intermediate-low resolution (S2 and S3 Figs) to increase power. The results also show that the two adult CVS samples are more similar to each other than to the fetal-like iPSC-CVPC.
(TIF)

**S5 Fig. Differential expression between CVS tissues.** (A, B) Volcano plots showing the differentially expressed genes between (A) iPSC-CVPC and adult arteria, and (B) adult heart and adult arteria. Genes that are not differentially expressed are shown in gray, differentially expressed genes with $\log_2$ ratio between -2 and 2 are in yellow, and genes with CVS tissue-specific expression (i.e. at least four-fold expression difference; $\log_2$ ratio $> 2$ or $< -2$) are shown in blue. (C, D) Barplots showing the $\log_2$ ratio between the mean expression in (C) iPSC-CVPC and adult arteria and (D) adult heart and adult arteria for 122 RBPs with a known motif [31–33]. RBPs shown in blue, maroon and green are iPSC-CVPC-specific, adult heart-specific and adult arteria-specific, respectively. All other differentially expressed RBPs (FDR $< 0.05$) are shown in yellow. These plots, in combination with Fig 1D, show that iPSC-CVPC have more overexpressed and specific RBPs than either adult heart or adult arteria, and while there is a greater number of overexpressed RBPs in adult arteria compared with adult heart, the two CVS tissues have similar numbers of tissue-specific RBPs expressed.
(TIFF)

**S6 Fig. RBPs differentially expressed between CVS tissues.** Boxplots showing gene expression levels (TPM) in iPSC-CVPC, adult heart and adult arteria for six RBPs involved in cardiac and muscle development. Four of these genes (*CELF1*, *HNRNPL*, *RBFOX2* and *RBM24*) have known functions during cardiac fetal development and were overexpressed in iPSC-CVPC compared with the two adult tissues, whereas the other two (*MBNL1* and *RBFOX1*) have adult-specific functions and were expressed at higher levels in adult arteria and adult heart, respectively, compared with iPSC-CVPC.
(TIFF)

**S7 Fig. Validation of differential isoform usage between fetal-like iPSC-CVPC and adult heart in cardiac marker genes.** The figure shows the differential isoform usage between iPSC-CVPC (blue in the central panel) and adult heart (purple) for four cardiac genes known to have well-established developmental stage-specific isoforms: *SCN5A*, *TNNT2*, *ABLIM1* and *TTN*. Left side of the figure represents the exon structure of each expressed isoform; the barplots in the middle show the mean isoform use (%) of each isoform in iPSC-CVPC (blue) and adult heart (atrium and ventricle, purple). Barplots on the right show the $\log_2$ ratio between the mean isoform use in iPSC-CVPC and adult heart. Red stars indicate significantly differentially expressed isoforms (FDR $< 0.05$). Dotted lines represent one-, two- and four-fold differences between iPSC-CVPC and adult. We found that three *SCN5A* isoforms, three *TNNT2* isoforms, three *ABLIM1* isoforms and four *TTN* isoforms were differentially expressed. Four of these isoforms (two for *SCN5A* and two for *TNNT2*) were tissue-specific, having at least a four-fold change difference between CVS tissues. To test if the isoforms expressed in the fetal-like iPSC-CVPC and in the adult heart reflect the known differences between fetal and adult, we further investigated their structure. The main differences between *SCN5A* fetal and adult isoforms are two mutually exclusive exons that encode for the voltage-sensing transmembrane domain and result in different electrophysiological properties: 6A (fetal) and 6B (adult) [102–104]. We confirmed that they were correctly expressed in the iPSC-CVPC- and adult-specific isoforms (S7A Fig). *TNNT2* isoforms differ at the 5' end, with the longer isoforms being fetal-specific and the shorter isoforms being expressed in the adult [66]. Of the three differentially

expressed isoforms, we observed that the two longer isoforms were iPSC-CVPC-specific, whereas the shorter was enriched in the adult heart (S7B Fig). ABLIM1 has four C-terminal LIM domains that play a role in its binding to actin [26]. The shorter isoforms, which only contains three LIM domains because of exon 11–12 skipping, are expressed earlier during development [25]. We confirmed that the two isoforms overexpressed in iPSC-CVPC do not express these two exons (S7C Fig). Likewise, *TTN* has multiple isoforms, which accomplish different functions during embryonic development and in the adult heart: the longer fetal isoforms are responsible for the low stiffness of the fetal myocardium, whereas the short adult isoforms are associated with increased passive myocardial stiffness and may play a role in the adjustment of the cardiac muscle to increased diastolic function during development [67]. Two of the shorter *TTN* isoforms were overexpressed in the adult heart (ENST00000460472.6_4: $p = 2.6 \times 10^{-19}$, ENST00000342992.10_4: $p = 6.1 \times 10^{-33}$), whereas two of the longer isoforms were overexpressed in iPSC-CVPC (ENST00000360870.9_4: $p = 9.0 \times 10^{-17}$, ENST00000470257.1_2: $p = 2.0 \times 10^{-13}$, S7D Fig). These observations confirm that cardiac genes with established fetal-specific and adult-specific isoforms, including *SCN5A*, *TTN*, *ABLIM1* and *TNNT2* [2,4,8,23–26,66–69], displayed the expected differential isoform usage in the iPSC-CVPC and adult heart.
(TIFF)

**S8 Fig. Fetal-like CVS tissue has more functionally active RBPs.** (A, C) Barplots showing the differential expression of 38 RBPs with experimentally determined (eCLIP) binding sites between (A) iPSC-CVPC and adult arteria and (C) adult heart and adult arteria. Colors represent $-\log_{10}$ p-values (S3 Table). Each row represents one RBP. (B, D) Plots showing the enrichment of genomic loci encoding (B) iPSC-CVPC-specific isoforms versus adult arteria-specific isoforms and (D) adult heart-specific isoforms versus adult arteria-specific isoforms for overlapping experimentally determined (eCLIP) RBP binding sites. Each line corresponds to an eCLIP experiment. The X axis corresponds to $\log_2$ ratio thresholds: (B) mean isoform usage in iPSC-CVPC/ mean isoform usage in adult arteria; and (D) mean isoform usage in adult heart/ mean isoform usage in adult arteria. Enrichment was calculated by comparing the proportion of genes with differentially expressed isoforms passing the $\log_2$ ratio thresholds described on the X axis (S5 Table) and overlapping eCLIP peaks against the proportion of genes without any differentially expressed isoforms (FDR > 0.05 for all the isoforms) overlapping eCLIP peaks. The thick lines represent the mean enrichments across all RBP experiments for iPSC-CVPC-specific isoforms (blue), adult heart-specific isoforms (purple) or adult arteria-specific isoforms (green). These plots show that genes with iPSC-CVPC-specific isoforms are more likely to overlap RBP binding sites than genes with adult arteria-specific isoforms; whereas adult heart-specific isoforms and adult arteria-specific isoforms are equally likely to overlap RBP binding sites.
(TIFF)

**S9 Fig. iPSC-CVPC-specific exons enriched for splice sites overlapping motifs of 122 RBPs.** Shown are the differences in the likelihood of the 100 bp upstream of the acceptor splice sites of iPSC-CVPC-specific, adult heart-specific and adult arteria-specific exons to harbor motifs of 122 RBPs. (A-C) Barplots showing the $\log_2$ ratio of the likelihood of each CVS tissue-specific splice site to harbor a RBP motif (iPSC-CVPC = blue, adult heart = red, adult arteria = green). (D-F) Volcano plots show the enrichment for each RBP (X axis, same as panels A-C) and its associated $-\log_{10}$(q-value) calculated using Homer *findMotifsGenome.pl* [98]. Q-values $< 1.0 \times 10^{-5}$ are shown as $1.0 \times 10^{-5}$. Colors are as shown in panels A-C.
(TIFF)

**S10 Fig. Clustering of FACs-sorted cardiac cells from *Tabula Muris*.** UMAP plot showing the clustering and cell types associated with each of the 4,365 cells obtained from *Tabula Muris*. Cell labels are as defined by the *Tabula Muris* Consortium [50].
(TIFF)

**S11 Fig. Principal components capture heterogeneity due to cellular composition and developmental-stage across 966 CVS samples.** (A) Heatmap showing the Pearson correlations between the top 5 principal components calculated on 2,000 most variable genes across the 966 bulk RNA-seq CVS samples, cell type proportion, and the tissue types of the 966 CVS samples. The top two principal components were correlated with developmental stage (R = 0.934) and cardiac muscle proportion (R = 0.864), respectively, indicating that gene expression levels are strongly associated with cellular heterogeneity and suggesting that the heterogeneity observed at high clustering resolution is likely a consequence of the variability in cell type proportions within each tissue. (B) Scatter plot showing the separation of fetal iPSC-CVPC (right) and adult heart and adult arteria (left) by the first principal component and the separation of samples with larger fraction of cardiac muscle (top/light blue) and those with smaller fraction (bottom/dark blue) by the second principal component. These results further highlight that developmental stage and cellular heterogeneity explain the most variability in the transcriptomes of the 966 CVS samples.
(TIF)

**S12 Fig. Associations between gene expression and cell type proportions.** Volcano plots showing the number of genes whose expression is associated with the proportion of indicated cell type. This analysis was performed on all 966 CVS samples using linear regression (*lm* function in R) between cell type proportions and gene expression levels. X axis represents the effect size and the Y axis represents p-values. All non-significant genes are colored in light gray (FDR > 0.05). Genes significantly associated with each cell type (FDR < 0.05 and effect size > 0) are colored and their number is reported at the bottom-right of each plot.
(TIFF)

**S13 Fig. Validating estimated cell-type proportions with scRNA-seq.** We used two scRNA-seq datasets from iPSC-CVPC and adult heart [14,52] to further characterize the cell type proportions estimated by CIBERSORT. Using scRNA-seq data, we performed differential expression analysis for each cell type and found that the ratio between the expression levels of each gene in a cell type compared with all the other cells was strongly correlated with the effect size of the associations between estimated cell type proportions determined by CIBERSORT deconvolution and gene expression across all cell types. (A-B) Plots showing the UMAP coordinates of (A) 32,026 iPSC-CVPC cells from eight iPSCORE samples (CM: cardiomyocytes; EPDC: epicardial-derived cells); and (B) 33,050 adult left ventricle cells. (C) Scatterplot showing the effect size (linear regression) of the association between gene expression and % of cardiac muscle in the 966 deconvoluted RNA-seq CVS samples (X axis) and the $\log_2$ ratio between the expression in CMs and EPDCs in the 32,026 iPSC-CVPC cells (Y axis). (D-G) Scatterplots showing the effect size of the association between gene expression and % of (D) cardiac muscle, (E) fibroblast, (F) endothelial and (G) smooth muscle in the 966 deconvoluted RNA-seq CVS samples (X axis) and the $\log_2$ ratio between gene expression in the indicated cell type and all other cells for the 33,050 adult left ventricle cells. Red shows highly expressed genes (expressed in at least 50% of iPSC-CVPC cells or 10% of adult cells). All the lowly expressed genes are shown in blue. Red and blue dashed lines are the regression lines for highly expressed genes and across all tested genes, respectively. In the bottom right corner of each plot, the correlation for the highly expressed genes is shown. In the top left corner, the

correlation calculated across all the genes is shown. These plots show that the associations between gene expression and the estimated cell type proportion in the bulk RNA-seq data are significantly strongly correlated with the expression differences observed in the scRNA-seq data, both in iPSC-CVPC and adult heart, indicating that cellular deconvolution and differential gene expression analysis reflect differences in cell composition between samples. (H) Scatterplot showing the difference between effect sizes of the association between cardiac muscle proportion and gene expression between iPSC-CVPC and adult heart (X axis) and the $\log_2$ ratio between average normalized expression in iPSC-CVPC and adult cardiac muscle cells. The strong positive correlation indicates that the tissue-specific associations in cardiac muscle between fetal-like iPSC-CVPC and adult heart reflect gene expression differences between fetal-like iPSC-CVPC and adult cardiac muscle cells observed using scRNA-seq. (TIFF)

**S14 Fig. Differential isoform expression: linear model versus ridge regression.** Scatterplots showing the differential isoform usage effect size without considering cell types (linear model, LM, X axis) and considering the cell type proportions as covariates (ridge regression, Y axis) for each pairwise comparison between CVS tissues: (A) iPSC-CVPC vs. adult heart and (B) adult heart vs. adult arteria. Isoforms differentially expressed in both analyses and that have the same effect size sign are shown in green; isoforms whose effect size has different signs between the LM and ridge regression are in red; isoforms that are differentially expressed only using the LM are shown in blue; and isoforms that are differentially expressed only when taking cell type proportions into account (ridge) are in cyan. (TIFF)

**S15 Fig. Gene expression associations with both CVS tissues and cell-type proportions.** Scatterplots showing differential gene expression effect size between each indicated pair of CVS tissues and the effect size of the association between gene expression and cell type proportion of cardiac muscle or smooth muscle cells. Genes whose expression are not associated with either CVS tissue or cell type proportion are shown in gray; genes significantly associated with both CVS tissue and cell type proportion are in green; genes associated only with cell type are in blue; and genes that are associated only with CVS tissue are in cyan. Overall, we observed a weak correlation between the effect size of the differential expression analysis between iPSC-CVPC and adult and the effect size of the association between gene expression and cardiac muscle proportion ($R^2$ = 0.00925, panel A). This correlation was stronger when comparing the effect size of the differential expression analysis against arteria ($R^2$ = 0.142 and $R^2$ = 0.383, respectively for iPSC-CVPC and adult heart, panels C, E), suggesting that, differences in cell type proportions drive the transcriptomic differences between iPSC-CVPC and adult arteria and between adult heart and adult arteria. Conversely, the differences that we observe between iPSC-CVPC and adult heart are likely due to changes in gene expression within the same cell type. (TIFF)

**S16 Fig. RBPs associated with CVS developmental stage and cardiac muscle proportions.** Scatterplots showing cardiac muscle proportions (X axis) and normalized expression in iPSC-CVPC and adult heart (Y axis) for the RBPs that were both differentially expressed between iPSC-CVPC and adult heart (FDR < 0.05) and positively associated with at least one cell type (effect size > 0 and FDR < 0.05). Dashed lines represent regression lines calculated on each CVS tissue. (TIFF)

**S1 Table. Description of subjects included in this study.** The table shows subject information for all the 491 individuals included in this study, including: subject ID; subject name; source (iPSCORE or GTEx); sex; age, height (inches); weight (pounds); BMI; and 20 genotype principal components (S1B Fig). Columns AC-AG represent the family information for each iPSCORE subject, as included in dbGaP (phs001325) as part of the iPSCORE Resource: "family ID" classifies the subject by family to identify related family members; "twin ID" identifies the dbGaP ID if the subject is a twin; "twin type" indicates the type of twin (MZ = monozygotic; DZ = dizygotic) if the subject is a twin; "father ID" and "mother ID" indicate the subject UUID of the respective parent of the subject if that parent is part of the iPSCORE resource. (CSV)

**S2 Table. Description of bulk RNA-seq samples.** Shown are the 966 bulk RNA-seq samples used in this study, including: source (iPSCORE or GTEx); subject ID; assay ID; SRA run ID for the GTEx RNA-seq samples downloaded from dbGaP; iPSCORE iPSC line identifier submitted to dbGaP (phs001325), iPSCORE unique differentiation identifier (UDID) assigned to all molecular data generated from same iPSC-CVPC differentiation; total number of reads; % uniquely mapped reads; % of mitochondrial reads, calculated as the number of reads mapping to mitochondrial genes divided by the total number of reads mapping to genes; tissue and organ associated with each sample (arteria, heart, iPSC-CVPC, aorta, coronary artery, atrial appendage or left ventricle: 0 = absent; 1 = present); 50 principal components calculated on the expression of 2,000 genes across the 966 CVS samples using Seurat; UMAP coordinates of each sample after clustering by Seurat; S phase and G2M and scores calculated using Seurat; pseudotime score of each sample using Monocle; cluster membership of each sample calculated using Seurat at four different resolutions (Figs 1A and 1B, S2 and S3); estimated cell type proportions deconvoluted using CIBERSORT; the last column ("Trimmed") indicates which iPSCORE samples had their read length trimmed to 75 bp to test whether different read lengths between iPSCORE and GTEx affect differential gene and isoform expression analyses. (CSV)

**S3 Table. Differential gene expression analysis.** Shown is the differential expression analysis between: iPSC-CVPC and adult heart; iPSC-CVPC and adult arteria; adult heart and adult arteria. For each gene, we report: gene ID; gene name; the tested tissues (tissue 1 and tissue 2); effect size, standard error, p-value and FDR correction (Bonferroni). Effect size > 0 corresponds to genes where the expression in tissue 1 is greater than tissue 2. Only differentially expressed genes are reported (FDR < 0.05). The differential expression analysis of all genes can be found at https://doi.org/10.6084/m9.figshare.13537343. (CSV)

**S4 Table. Functional enrichment analysis (genes).** The table shows the functional enrichment analysis for genes differentially expressed between each pair of CVS tissues (S3 Table). For each gene set, we report: the tested tissues (tissue 1 and tissue 2); the gene set collection, as defined by MSigDB, the gene set name and its URL; the number of tested genes in the gene set; the average effect size for all the tested genes in the gene set and for all the other expressed genes; p-value (t-test); and FDR (Benjamini-Hochberg). Only significant gene sets are reported (FDR < 0.05). The differential expression analysis of gene sets can be found at https://doi.org/10.6084/m9.figshare.13537343. (CSV)

**S5 Table. Differential isoform expression analysis.** Shown is the differential isoform expression analysis between: iPSC-CVPC and adult heart; iPSC-CVPC and adult arteria; adult heart and adult arteria. For each isoform, we report: isoform ID, gene ID, gene name; the tested

tissues (tissue 1 and tissue 2); effect size, standard error, p-value and FDR correction (Bonferroni); mean isoform use in each of the two tested tissues and the $\log_2$ ratio between them; whether the isoform is differentially expressed (FDR < 0.05). Effect size > 0 corresponds to isoforms where the expression in tissue 1 is greater than tissue 2. Only CVS tissue-specific isoforms are reported (FDR < 0.05 and absolute value of $\log_2$ ratio >2). The differential expression analysis of all isoforms can be found at https://doi.org/10.6084/m9.figshare.13537343. (CSV)

**S6 Table. Functional enrichment analysis (isoforms).** The table shows the functional enrichment analysis of genes that have at least one CVS tissue-specific isoform, compared with genes that do not have differentially expressed isoforms between each pair of CVS tissues (S5 Table). For each gene set, we report: the tested tissues (tissue 1 and tissue 2); the gene set collection, as defined by MSigDB, the gene set name and its URL; the number of tested genes in the gene set; the enrichment calculated using the *estimate* parameter in the *fisher.test* function in R; p-value (Fisher's exact test); and FDR (Benjamini-Hochberg). Only significant gene sets are reported (FDR < 0.05). The differential expression analysis of gene sets can be found at https://doi.org/10.6084/m9.figshare.13537343. (CSV)

**S7 Table. Overlap of each CVS tissue-specific exon with protein domains.** The table shows the overlap between each CVS tissue-specific exon and protein domains obtained from the Prot2HG database [44]. For each CVS tissue-specific exon, we report: the gene ID, gene name, transcript ID, exon ID; the tested tissues ("iPSC-CVPC vs. heart", "iPSC-CVPC vs. arteria" or "heart vs. arteria"); the tested CVS tissue the exon is expressed in, and the overlapping protein domain of the exon. Exons that do not overlap known protein domains have the last column empty. (CSV)

**S8 Table. Enrichment of CVS tissue-specific isoforms for RBP binding sites.** For each pairwise comparison between CVS tissues (columns A and B), the table shows the differences in the overlap with each eCLIP experiment [31] between the gene bodies of isoforms specific to one of the two tested CVS tissues and the gene bodies of isoforms that are not differentially expressed. Enrichments were performed using increasingly stringent thresholds for determining CVS tissue-specific isoforms, based on the $\log_2$ ratio between the mean isoform usage in the two tested CVS tissues. The table shows: 1) the two tested CVS tissues (columns A, B); the tested RBP, its associated eCLIP experiment and the cell line in which the eCLIP experiment was performed (columns C-E); the $\log_2$ ratio threshold (zero to six, 0.5 increments); and the results obtained from Fisher's exact test (*fisher.test* function in R), including the estimate (odds ratio), the $\log_2$ of the estimate, and the p-value. Values displayed in this table were used to build Figs 3B and S8. (CSV)

**S9 Table. Enrichment of CVS tissue-specific exons for RBP motifs.** Tables showing the RBP motif enrichment results obtained using Homer *findMotifsGenome.pl* with CVS tissue-specific exons and 262 motifs for the 122 RBPs in the RNA binding gene set (GO:0003723). The table shows: RBP name and motif ID (columns A, B); the tested tissues (columns C, D); $\log_2$ ratio between the two tissues calculated by Homer (column E); p-value and q-value calculated by Homer (columns F, G). (XLSX)

**S10 Table. Enrichment of CVS tissue-specific exons for canonical splice sites.** For each pairwise comparison between CVS tissues, (column A, B: iPSC-CVPC vs. adult heart, iPSC-CVPC vs. adult arteria and adult heart vs. adult arteria), the table shows the differences in the probability of each base pair occurring in the 100 bp upstream of the splice acceptor site and the 100 bp downstream of the splice donor site. The table shows: the two tested CVS tissues (columns A, B); which splice site is tested (donor or acceptor, column C); the tested nucleotide (column D); the position relative to the exon start (negative values, upstream of the splice acceptor site) or end (positive values, downstream of the splice donor site) (column E); and the results obtained from Fisher's exact test (*fisher.test* function in R), including the estimate (odds ratio, column F), its 95% confidence interval (column G, H) and the p-value (column I). In Fig 3C–3I, only the tests associated with the most common nucleotide at each position are shown. Fig 3D, 3F and 3H shows the estimate from Fisher's exact test and its 95% confidence interval; whereas Fig 3E, 3G and 3I shows the p-values calculated with Fisher's exact test.
(CSV)

**S11 Table. Marker genes for cardiac cell types in *Tabula Muris*.** Shown are marker genes identified for each cluster using the *FindAllMarkers* function in Seurat. Column A and B report the gene name and tested cell type, respectively; column C represents the % of cells labeled as the cell type described in column B that express the gene; column D represents the % of cells in all other cell types that express the same gene; column E shows the $\log_2$ fold change between the average expression level in the cell type described in column B and all other cells; column F and G represent the p-value (Wilcoxon rank sum test) and FDR-adjusted p-value (Bonferroni). All values described in this table were calculated using the *FindAllMarkers* function in Seurat with parameters *min.pct = 0.2*, *logfc.threshold = 0.1*. Only genes with FDR < 0.1 are shown. If a gene is significantly overexpressed in more than one cell type, it is reported multiple times.
(CSV)

**S12 Table. Expression matrix input to CIBERSORT.** Shown is the average expression of all marker genes described in S11 Table. Expression levels were obtained using the Seurat function *AverageExpression*. This table was used as input for CIBERSORT. CIBERSORT results are reported in S2 Table with all the variables used as covariates for the differential expression analysis.
(CSV)

**S13 Table. Differential gene and isoform expression analysis (cell types).** Shown are the associations between the expression of each expressed gene and isoform and cell type proportions in all bulk RNA-seq CVS samples. For each gene and isoform, we report: isoform ID, gene ID, gene name; the cell type whose proportion is tested for association with expression; effect size, standard error, p-value (linear regression: *lm* function in R), FDR correction (Bonferroni); whether the expression of the gene or isoform is associated with cell type proportion (FDR < 0.05 and effect size > 0). Only differentially expressed genes and isoforms are reported (FDR < 0.05). The differential expression analysis of all genes and isoforms can be found at https://doi.org/10.6084/m9.figshare.13537343.
(CSV)

**S14 Table. Functional enrichment analysis (cell types).** The table shows the functional enrichment analysis for genes associated with each cell type. The table is organized as S4 Table, with one difference: cell type, rather than "tissue 1" and "tissue 2" is reported. For each cell type, we determined the association (effect size) between its proportion and the expression of each gene, and used the effect sizes of all expressed genes as input for gene set enrichment

analysis (GSEA). We observed that the most significantly enriched gene sets corresponded to the main function associated with each cell type, including mitochondrial and cell respiration functions for cardiac muscle, immune response for immune cells, cytoskeleton and actin binding for smooth muscle. Only significant gene sets are reported (FDR < 0.05). The differential expression analysis of gene sets can be found at https://doi.org/10.6084/m9.figshare.13537343. (CSV)

**S15 Table. Differential gene and isoform expression using cell type proportions as covariates.** Shown is the differential expression analysis for both genes and isoforms performed using ridge regression and cell type proportions as covariates between: iPSC-CVPC and adult heart; iPSC-CVPC and adult arteria; adult heart and adult arteria. For each gene and isoform, we report: isoform ID, gene ID and gene name; the tested tissues (tissue 1 and tissue 2); the covariate, including "tissue", which represents the differential expression between tissue 1 and tissue 2, and each of the cell types; effect size, standard error, p-value and FDR correction (Bonferroni). Only significant associations (FDR < 0.05) are reported. All tests can be found at https://doi.org/10.6084/m9.figshare.13537343. The associations between fibroblast and immune cell type proportion and developmental stage (iPSC-CVPC vs. adult heart) are likely noisy, because these two cell types were not present in iPSC-CVPCs. Therefore the genes and isoforms associated with these two cell types were not considered in the count of genes and isoforms associated with both developmental stage and cell types in the ridge regression analysis. (CSV)

**S16 Table. PCA of heart failure, iPSC-CVPC, adult heart and adult arteria bulk RNA-seq samples.** The table shows sample information for each of the 30-heart failure bulk RNA-seq samples (15 pairs of matched pre- and post-LVAD samples from GSE46224) and the principal components coordinates of these samples, as well as the 966 iPSCORE and GTEx CVS samples. Columns A-C show the sample ID, tissue and study for all the 996 analyzed samples. Columns D-F show subject ID, age, sex and LVAD status (pre- or post-LVAD) for each of the 30 heart failure samples. The subsequent columns show the PCA coordinates of all the samples based on gene expression. (CSV)

**S17 Table. Cell proportion differences between heart failure and healthy CVS tissues.** The table shows the cell proportion differences between the two heart failure sample sets (pre- and post-LVAD) and the three healthy CVS tissues (iPSC-CVPC, adult heart and adult arteria). For each pair of tissues (columns A and B) and cell type (column C), the mean cell type proportion in each of the two tissues is shown (columns D and E), as well as the p-value (t-test, column F) and its FDR correction (Benjamini-Hochberg's method, column G). We found that pre-LVAD samples had a significantly higher proportion of myofibroblasts than post-LVAD samples (2.1%, compared with 1.2%, p = 0.0010, t-test). All the other cell type proportion differences between pre- and post-LVAD cell type proportions were not significant after FDR correction. (CSV)

**S18 Table. Differential expression between heart failure and healthy CVS tissues.** The table shows the results from differential expression analysis of genes and isoforms between pre-LVAD samples and the three healthy CVS tissues (iPSC-CVPC, adult heart and adult arteria). Columns A-D show gene and transcript information, including: transcript ID (only for isoforms; this column is left blank for genes), gene ID, gene name, and analysis type (gene or isoform). Columns E-F show the pairs of tested tissues. Columns G-J show the effect size, its standard error, p-value and FDR correction (Bonferroni's method). FDR correction was

performed independently on genes and isoforms. All genes and isoforms with FDR < 0.05 were considered as significant and shown in this table. All tests can be found at https://doi.org/10.6084/m9.figshare.13537343.
(CSV)

**S19 Table. Genes and isoforms whose expression is associated with read length.** The table shows the genes and isoforms that had significant expression level differences between 50 iPSCORE RNA-seq samples with two different read lengths (150 bp vs. 75 bp trimmed). For each gene or isoform, shown are the gene or isoform ID, p-value (paired t-test), FDR (Bonferroni-corrected p-value), source (gene or isoform), gene ID if the source is isoform, and whether the gene or isoform is blacklisted. We blacklisted genes whose expression was significantly different between the 50 samples at the two read lengths or that had isoforms whose expression was significantly different. This resulted in the removal of 1,059 genes and their 8,999 isoforms.
(CSV)

## Acknowledgments

We thank Patrick T. Ellinor and Mark Chaffin for sharing the adult scRNA-seq data prior to release (dbGaP phs001539).

## Author Contributions

**Conceptualization:** Matteo D'Antonio, Kelly A. Frazer.

**Data curation:** Jennifer P. Nguyen, Timothy D. Arthur, Hiroko Matsui.

**Formal analysis:** Matteo D'Antonio, Jennifer P. Nguyen, Timothy D. Arthur, Hiroko Matsui, Margaret K. R. Donovan.

**Funding acquisition:** Kelly A. Frazer.

**Investigation:** Matteo D'Antonio.

**Methodology:** Matteo D'Antonio, Jennifer P. Nguyen, Timothy D. Arthur, Hiroko Matsui, Margaret K. R. Donovan, Agnieszka D'Antonio-Chronowska.

**Project administration:** Kelly A. Frazer.

**Resources:** Agnieszka D'Antonio-Chronowska.

**Supervision:** Agnieszka D'Antonio-Chronowska, Kelly A. Frazer.

**Writing – original draft:** Matteo D'Antonio, Kelly A. Frazer.

**Writing – review & editing:** Matteo D'Antonio, Kelly A. Frazer.

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
