## [Decision Letter · Decision Letter 0]

6 Sep 2021

Dear Dr. Frazer,

Thank you very much for submitting your manuscript "Reactivation of fetal stage RNA-binding proteins and isoforms in heart failure" for consideration at PLOS Computational Biology.

As with all papers reviewed by the journal, your manuscript was reviewed by members of the editorial board and by several independent reviewers. In light of the reviews (below this email), we would like to invite the resubmission of a significantly-revised version that takes into account the reviewers' comments.

We cannot make any decision about publication until we have seen the revised manuscript and your response to the reviewers' comments. Your revised manuscript is also likely to be sent to reviewers for further evaluation.

Sincerely,

Zhaolei Zhang

Associate Editor

PLOS Computational Biology

Jian Ma

Deputy Editor

PLOS Computational Biology

Reviewer's Responses to Questions

**Comments to the Authors:**

Reviewer #1: The authors integrated 996 RNA-seq samples from heart failure, heathy heart and arteria as well as fetal-like CVS tissues and revealed that RNA-binding proteins are highly overexpressed in fetal-like compared with health adult and are reactivated in heart failure. Also, they showed that overexpression of RBPs in heart failure is associated with a transcriptome-wide isoform switch, which could be used as novel therapeutics for heart failure.

Overall, the manuscript is well written and easy to follow. The description of the methods and data sources is very clear. All calculations and use of statistics throughout the manuscript were properly carried out. I have no major concerns of the approach or the conclusions.

Minor Comments:

Figure 6B: The colors of LVAD are a little confused, could you change the colors of the arrows?

Figure 5D-G: Could you add the p-values on the figures so that readers could easily catch the information by only looking at the figures.

Reviewer #2: This research work focuses on investigating differential gene expressions and alternative splicings related to heart failure. Bulk RNAseq data of a health patient cohort matched with samples of induced pluripotent stem cells from cardiovascular progenitor cells (iPSC-CVPC) were generated and analyzed to identify differential expressed genes and isoforms of alternative use between iPSC-CVPC samples and adult samples. The analysis revealed significant differential expressions with stage specific isoforms in RNA binding proteins (RBPs). Further comparison with another cohort of heart failure patient data reveals fetal-like expressions of RBPs and informs in these heart failure samples. While the study uses a novel approach to investigate an important research question, the overall study design and the data analysis seem to need some more improvements as outlined below,

Major:

1. One strength of the study is the integration of multiple sets of samples and datasets to make the investigation possible. However, one caveat is the possible technical biases and batch effects among the gene expression datasets. The sample separations in Figure 1(A) could be due to tissue/developmental stage specificity or technical bias from different preparation of iPSC-CVPC and iPSCORE samples. The linear regression model with additive factors cannot distinguish tissue type from data/sample batch. It might be a good idea to utilize the paired samples between iPSC-CVPCs and adult hear/arteria samples to improve the reliability of the analysis. Given the large number of differentially expressed genes detected in the analysis, the results in Figure 1 and Figure 2 should be at least further validated with the paired samples. Without distinguishing tissue type and such possible technical bias or batch effect, the results are not convincing. This is a major limitation of the current analysis.

2. Given the more biological focus of this study, it is somewhat expected that the findings should be further validated with additional laboratory experiments. For example, the alternative splicing and differential expressions in some specific RBPs shown in Figure 2 can be further validated with PCR or protein staining. Without such validation (even a small scale one), the findings still remain quite hypothetical.

3. In the results on page 6 and Figure 2, the analysis of isoform only focuses on the proportions. It is also important to show differential absolute expression or the genes/isoforms, given the selected examples of RBPs in Figure 1 and Figure 2 are quite different. When the gene expression is lower, the isoform quantification is often less reliable. It is important to also visualize the expressions in addition to the proportions.

4. The organization of the article especially the method section can still be improved. Many different analyses are described with very sparse detail. There are three major analysis in the manuscript, differential gene expressions and alternative splicing analysis, functional motif/exon enrichment, and cell type decomposition. The specific analysis steps can be better integrated and explained together for each of the major analysis.

Minor:

1. In the results on top of page 5, it is not stated how many samples are determined heart and how many are arteria. Neither in Figure 1 and Figure S2.

2. The results in Figure 3B and Figure S8 are confusing. Especially, it seems the eCLIP peak enrichment/depletion switches when different log2 thresholds are used to pick the candidate isoforms of each gene. More explanation and justification are needed to support the conclusion.

3. It is unclear if all the analyses are done after the integration of iPSCORE and GTEx datasets as described on page 14. This step is unnecessary for the analysis between IPSC-CVPC and iPSCORE samples. Please clarify.

4. The step of removing genes and isoforms for the correction of read length bias is somewhat strange. Isn’t read length bias systematic rather than gene or isoform specific? The correction does not seem to accomplish the the goal.

5. In “Defining RBPs”, are the RBPs already given from the sources or identified by matching the PWMs again the genes in this study?

6. Please provide more detail (statistics) of eCLIP peak data. How was the peak called?

Reviewer #3: This manuscript performed systematic analyses of published datasets (GTEx, iPSCORE, heart failure bulk RNA-seq, Tabula Muris) and investigated the role of RNA binding protein and splicing in CVS development and heart failure. I found that the manuscript is generally interesting, and the analyses are comprehensive. Still, I would prefer more details/clarifications of the methods and results to better facilitate readers’ understanding of this manuscript.

In the abstract, the authors state that ‘Comparison of the expression profiles revealed that RNA-binding proteins (RBPs) are highly overexpressed in fetal-like compared with healthy adult and are reactivated in heart failure, which results in expression of thousands fetal-specific isoforms’ I am not sure whether the manuscript has enough data to support the causality.

More detailed information should be provided regarding how the differential expression analysis and the isoform analysis were performed. The methods section should provide more information regarding this.

In Figure 3A-B, are there differences between the 33 RBPs which were overexpressed in iPSC-CVPC and the six which were not differentially expressed?

Interestingly, the heart failure sample has fetal-like expression patterns. How about other types of cardiovascular disease? I assume this observation will be more relevant to the disease related to cardiovascular development versus the others.

It is unclear how the motif analysis for the 122 RBPs (Figure S9) was conducted. Please provide more information to clarify this analysis. Was the analysis performed on all the iPSC-CVPC-specific, adult heart-specific, and adult arteria-specific exons using the motif of the 122 RBPs? Which motifs (source) were used, and how the motif search was performed?

**Have the authors made all data and (if applicable) computational code underlying the findings in their manuscript fully available?**

Reviewer #1: Yes

Reviewer #2: Yes

Reviewer #3: **No: **Data are available as suggested by the authors, but codes are not (not sure whether it is a must for PLOS computational biology)

PLOS authors have the option to publish the peer review history of their article (what does this mean?). If published, this will include your full peer review and any attached files.

Reviewer #1: No

Reviewer #2: No

Reviewer #3: No
---

## [Decision Letter · Decision Letter 1]

4 Jan 2022

Dear Dr. Frazer,

Thank you very much for submitting your manuscript "In heart failure reactivation of RNA-binding proteins is associated with the expression of 1,523 fetal-specific isoforms" for consideration at PLOS Computational Biology. As with all papers reviewed by the journal, your manuscript was reviewed by members of the editorial board and by several independent reviewers. The reviewers appreciated the attention to an important topic. Based on the reviews, we are likely to accept this manuscript for publication, providing that you modify the manuscript according to the review recommendations.

Sincerely,

Zhaolei Zhang

Associate Editor

PLOS Computational Biology

Jian Ma

Deputy Editor

PLOS Computational Biology

[LINK]

Reviewer's Responses to Questions

**Comments to the Authors:**

Reviewer #1: Thank you for addressing my concerns, I have no further questions.

Reviewer #2: The authors addressed most of the comments but some concerns regarding the analysis and validation of the differential isoform expressions remain. Specifically,

1. What is the relation between differential gene expression and differential isoform proportion expression? It is understood that only the genes that are considered expressed are used for differential isoform proportion analysis. It also could be true that the isoforms' proportion is more relevant than absolute expression. However, this does not dismiss the importance of analyzing the differential proportions with the context of the absolute gene expression. It will be at least useful to find out whether differential expressed genes also tent to have differential isoform proportion or vice versa etc.? Given the large number of differentially expressed isoforms (19270), more detailed analysis is necessary.

2. It is also understood that there are evidence from the literature supporting the findings in this study. However, it seems none of the evidences is at the isoform level. In Figure 2B-E and Figure S7, the results are from the RNAseq data used for the discovery, which are not validation. Since the discoveries of these isoforms is part of the important contribution of the work, it will be useful to find literature evidence/existing data/new experiments directly support the discoveries. Maybe, some other small scale RNAseq data or protein profiling/staining from other studies can be used for the validation?

Overall, the study is solid but the analysis and the validation of the findings of the isoforms are still lacking.

Reviewer #3: The authors have addressed most of my questions. I recommend acceptance of the paper.

**Have the authors made all data and (if applicable) computational code underlying the findings in their manuscript fully available?**

Reviewer #1: Yes

Reviewer #2: **No: **There is no new software developed in the study. All the sequencing data are publicly available but there is sharing of any source code or processed data.

Reviewer #3: Yes

PLOS authors have the option to publish the peer review history of their article (what does this mean?). If published, this will include your full peer review and any attached files.

Reviewer #1: No

Reviewer #2: No

Reviewer #3: No

Figure Files:

Data Requirements:

Reproducibility:

References:

---

## [Decision Letter · Decision Letter 2]

10 Feb 2022

Dear Dr. Frazer,

We are pleased to inform you that your manuscript 'In heart failure reactivation of RNA-binding proteins is associated with the expression of 1,523 fetal-specific isoforms' has been provisionally accepted for publication in PLOS Computational Biology.

Best regards,

Zhaolei Zhang

Associate Editor

PLOS Computational Biology

Jian Ma

Deputy Editor

PLOS Computational Biology

Reviewer's Responses to Questions

**Comments to the Authors:**

Reviewer #2: This revision addressed all my comments.

**Have the authors made all data and (if applicable) computational code underlying the findings in their manuscript fully available?**

Reviewer #2: Yes

PLOS authors have the option to publish the peer review history of their article (what does this mean?). If published, this will include your full peer review and any attached files.

Reviewer #2: **Yes: **Rui Kuang

---

## [Editor Report · Acceptance letter]

24 Feb 2022

PCOMPBIOL-D-21-01231R2 

In heart failure reactivation of RNA-binding proteins is associated with the expression of 1,523 fetal-specific isoforms

Dear Dr Frazer,

I am pleased to inform you that your manuscript has been formally accepted for publication in PLOS Computational Biology. Your manuscript is now with our production department and you will be notified of the publication date in due course.

With kind regards,

Zsofia Freund
